# LONG-TERM IMPACTS OF MODEL RETRAINING WITH STRATEGIC FEEDBACK

## ABSTRACT

When machine learning (ML) models need to be frequently retrained, it is often too expensive to obtain *human-annotated* samples, so recent ML models have started to label samples by themselves. This paper studies a setting where an ML model is retrained (with *human* and *model-annotated* samples) over time to make decisions about a sequence of *strategic* human agents who can adapt their behaviors in response to the most recent ML model. We aim to investigate what happens when *model-annotated* data are generated under the agents' strategic feedback and how the models retrained with such data can be affected. Specifically, we first formalize the interactions between agents and the ML system and then analyze how the agents and ML models evolve under such dynamic interactions. We find that as the model gets retrained, agents are increasingly likely to receive positive decisions, whereas the proportion of agents with positive labels may decrease over time. We thus propose an approach to stabilize the dynamics and show how this method can further be leveraged to enhance algorithmic fairness when agents come from multiple social groups. Experiments on synthetic/semi-synthetic and real data validate the theoretical findings.

## 1 INTRODUCTION

As machine learning (ML) is widely used to automate human-related decisions (e.g., in lending, hiring, college admission), there is a growing concern that these decisions are vulnerable to human strategic behaviors. With the knowledge of decision policy, humans may adapt their behavior strategically in response to ML models, e.g., by changing their features at costs to receive favorable outcomes. A line of research called *Strategic Classification* studies such problems by formulating mathematical models to characterize strategic interactions and developing algorithms robust to strategic behavior (Hardt et al., 2016; Levanon

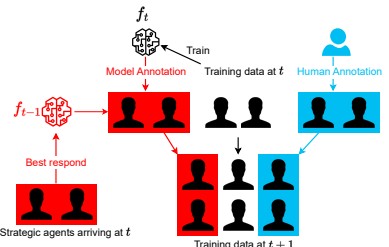

Figure 1: Illustration of model retraining with strategic feedback from $t$ to $t + 1$

& Rosenfeld, 2022). Among the existing works, most studies focus on one-time deployment where an ML model is trained and applied to a fixed population *once*.

However, practical ML systems often need to be retrained periodically to ensure high performance on the current population. As the ML model gets updated, human behaviors also change accordingly. To prevent the potential adverse outcomes, it is critical to understand how the strategic population is affected by the model retraining process. Traditionally, the training data used for retraining models can be constructed manually from *human-annotated* dataset (e.g., ImageNet). However, acquiring a large amount of *human-annotated* training samples can be highly difficult and even infeasible, especially in human-related applications (e.g., in automated hiring where an ML model is used to identify qualified applicants, even an experienced interviewer needs time to label an applicant).

Motivated by a recent practice of *automating* data annotation for retraining large-scale ML models (Taori & Hashimoto, 2023; Adam et al., 2022), we study strategic classification in a sequential framework where an ML model is periodically retrained by a decision-maker with both *human* and *model-annotated* samples. The updated models are deployed sequentially on agents who may change

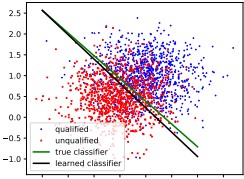 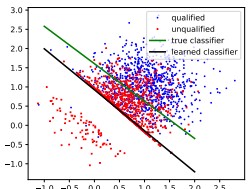 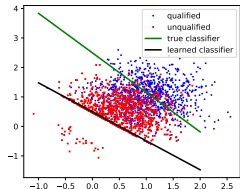

Figure 2: Evolution of the student distribution and ML model at $t = 0$ (left), $t = 5$ (middle), and $t = 14$ (right): each student has two features. At each time, a classifier is retrained with both human and model-annotated samples, and students best respond to be admitted as illustrated in Fig. 1. Over time, the learned classifier (black lines) deviates from ground truth (green lines).

their features to receive favorable outcomes. Specifically, we consider practical settings where: (i) the decision-maker can only label a limited number of *human-annotated* samples by itself, and has to use the current classifier to produce *model-annotated* samples for future retraining; (ii) the strategic agents need time to adapt their behaviors (Zrnic et al., 2021) and they best respond based on the previous model; (iii) feature changes caused by agents' best responses can genuinely change their underlying labels (Kleinberg & Raghavan, 2020), and feature-label relationship is fixed over time. Because the ML model affects agent behavior and such strategic feedback is further captured when retraining the future model, both the model and agents change over time. However, it remains unclear how the two evolve under such dynamics and what long-term effects one may have on the other.

In this paper, we examine the evolution of the ML model and the agent data distribution after they best respond. In particular, we ask: 1) How is the agent population reshaped over time when the model is retrained with strategic feedback? 2) How is the ML system affected by the agent's strategic response? 3) If agents come from multiple social groups, how can model retraining further impact algorithmic fairness? 4) What happens if the *human-annotated* samples have a systematic bias?

To further illustrate our problem, consider an example of college admission where new students from a population apply each year. In the $t$-th year, an ML model $f_t$ is learned from a training dataset $\mathcal{S}_t$ and used to make admission decisions. For students who apply in the $(t + 1)$-th year, they will best respond to the model $f_t$ in the previous year (e.g., preparing the application package in a way that maximizes the chance of getting admitted). Meanwhile, the college retrains the classifier $f_{t+1}$ using a new training dataset $\mathcal{S}_{t+1}$, which consists of previous training data $\mathcal{S}_t$, new *human-annotated* samples, and new *model-annotated* samples (i.e., previous applicants annotated by the most recent model $f_t$). This retrained model $f_{t+1}$ is then used to make admission decisions in the $(t + 1)$-th year. This process continues over time and we demonstrate how the training dataset $\mathcal{S}_t$ is updated in Fig. 1. Under such dynamics, both the ML system and the strategic population change over time and may lead to unexpected long-term consequences. An illustrating example is given in Fig. 2.

Compared to prior studies on *strategic classification*, we go beyond one-shot settings to study the long-term impacts of retraining in a sequential framework. Instead of assuming labels are available while retraining, we consider more practical scenarios with *model-annotated* samples. Although the risk of using *model-annotated* samples to retrain models has been highlighted in some existing works (Taori & Hashimoto, 2023), ours is the first to incorporate strategic feedback from human agents. More related works are discussed in App. C. Our contributions are summarized as follows:

1. We formulate the problem of model retraining with strategic feedback and qualitatively analyze the sources influencing the system dynamics (Sec. 2).
2. We theoretically characterize the evolution of the expectation of *acceptance rate* (i.e., the proportion of agents receiving positive classifications), *qualification rate* (i.e., the proportion of agents with positive labels), and the *classifier bias* (i.e., the discrepancy between acceptance rate and qualification rate) under the retraining process. We show that the acceptance rate increases over time under the retraining process, while the actual qualification rate may decrease under certain conditions. The dynamics of *classifier bias* are more complex depending on the systematic bias of *human-annotated* samples. Finally, we propose an approach to stabilize the dynamics (Sec. 3).
3. We consider settings where agents come from multiple social groups and investigate how inter-group fairness can be affected by the model retraining process; we also propose an *early stopping* mechanism to promote fairness (Sec. 4).
4. We conduct experiments on synthetic/semi-synthetic and real data to verify the theoretical results and test their robustness (Sec. 5, App. E, App. F).

## 2 PROBLEM FORMULATION

Consider a population of agents who are subject to certain machine learning decisions (e.g., admission/hiring decisions) and join the decision-making system in sequence. Each agent has observable continuous features $X \in \mathbb{R}^d$ and a hidden binary label $Y \in \{0, 1\}$ indicating its qualification state ("1" being qualified and "0" being unqualified). Let $P_{XY}$ be the joint distribution of $(X, Y)$ which is fixed over time, and $P_X$, $P_{Y|X}$ be the corresponding marginal and conditional distributions. $P_X$, $P_{Y|X}$ are continuous with non-zero probability mass everywhere in their domain. For agents who join the system at time $t$, the decision-maker makes decisions about them using a classifier $f_t : \mathbb{R}^d \to \{0, 1\}$. In this paper, we consider practical settings that the decision-maker does not know $P_{XY}$ and can only learn $f_t$ from the training dataset at $t$ (Guldogan et al., 2022).

**The agent's best response.** Agents who join the system at time $t$ can adapt their behaviors based on the latest classifier $f_{t-1}$ and change their features $X$ strategically. We denote the resulting data distribution as $P_{XY}^t$. Specifically, given original features $X = x$, agents have incentives to change their features at costs to receive positive classification outcomes, i.e., by maximizing utility

$$x_t = \max_z \ \{f_{t-1}(z) - c(x, z)\} \tag{1}$$

where distance function $c(x, z) \geqslant 0$ measures the cost for an agent to change features from $x$ to $z$. In this paper, we consider $c(x, z) = (z - x)^T B(z - x)$ for some $d \times d$ positive semidefinite matrix $B$, allowing heterogeneous costs for different features. After agents best respond, the agent data distribution changes from $P_{XY}$ to $P_{XY}^t$. In this paper, we term $P_{XY}$ agents' *prior-best-response* distribution and $P_{XY}^t$ agents' *post-best-response* distribution. We consider natural settings that (i) the agents' responses are *delayed*: they act based on the latest classifier $f_{t-1}$ they are aware of, not the one they receive; (ii) agents' behaviors are benign and cause the actual labels to change, so the relationship between features and label $P_{Y|X}^t = P_{Y|X}$ does not change (Guldogan et al., 2022).

**Human-annotated samples and systematic bias.** At each round $t$, we assume the decision-maker can draw a limited number of unlabeled samples from the prior-best-response distribution $P_X$. Note that the human annotation process is independent of the decision-making process. At $t$, each agent is classified by the model $f_t$ and best responds to $f_{t-1}$, the *decision-maker* never confuses the agents by simultaneously using human experts to label agents. Instead, human experts never participate in the interaction and human annotation is another process for the decision-maker to obtain additional information about the whole population (e.g., by first acquiring data from public datasets or third parties, and then labeling them to recover the population distribution). We may also consider the situation where *human-annotated* samples at $t$ are drawn from *post-best-response* distribution $P_X^t$, the discussion is in App. D.1. With some prior knowledge (possibly biased), the decision-maker can annotate these features and generate *human-annotated* samples $\mathcal{S}_{o,t}$. We assume the quality of human annotations is consistent, so $\mathcal{S}_{o,t}$ at any $t$ is drawn from a fixed probability distribution $D_{XY}^o$ with marginal distribution $D_X^o = P_X$. Because human annotations may not be the same as true labels, $D_{Y|X}^o$ can be biased compared to $P_{Y|X}$. We define such difference as the decision-maker's *systematic bias*, formally stated below.

**Definition 2.1** (Systematic bias). Let $\mu(D^o, P) = \mathbb{E}_{x \sim P_X}[D_{Y|X}^o(1|x) - P_{Y|X}(1|x)]$. The decision-maker has a systematic bias if $\mu(D^o, P) > 0$ (overestimation) or $\mu(D^o, P) < 0$ (underestimation).

Def. 2.1 implies that the decision-maker has a systematic bias when it labels a larger (or smaller) proportion of agents as qualified compared to the ground truth. Depending on the applications, the systematic bias may or may not exist and we study both scenarios in the paper.

**Model-annotated samples.** In addition to human-annotated samples, the decision-maker at each round $t$ can also leverage the most recent classifier $f_{t-1}$ to generate *model-annotated* samples for training the classifier $f_t$. Specifically, let $\{x_{t-1}^i\}_{i=1}^N$ be $N$ post-best-response features (equation 1) acquired from agents coming at $t-1$, the decision-maker uses $f_{t-1}$ to annotate the samples and obtain *model-annotated* samples $\mathcal{S}_{m,t-1} = \{x_{t-1}^i, f_{t-1}(x_{t-1}^i)\}_{i=1}^N$. Both human and model-annotated samples can be used to retrain the classifier at $t$.

**Classifier's retraining process.** With the human and model-annotated samples introduced above, we next introduce how the model is retrained by the decision-maker over time. Denote the training set at $t$ as $\mathcal{S}_t$. Initially, the decision-maker trains $f_0$ with a *human-annotated* training dataset $\mathcal{S}_0 = \mathcal{S}_{o,0}$. Then the decision-maker updates $f_t$ every round to make decisions about agents. The decision-maker

learns $f_t \in \mathcal{F}$ using empirical risk minimization (ERM) with training dataset $\mathcal{S}_t$. Similar to studies in strategic classification (Eilat et al., 2022), we consider linear hypothesis class $\mathcal{F}$. At each round $t \geqslant 1$, $\mathcal{S}_t$ consists of three components: existing training samples $\mathcal{S}_{t-1}$, $N$ new *model-annotated* and $K$ new *human-annotated* samples, i.e.,

$$\mathcal{S}_t = \mathcal{S}_{t-1} \cup \mathcal{S}_{m,t-1} \cup \mathcal{S}_{o,t-1}, \ \ \forall t \geqslant 1 \tag{2}$$

Since annotating agents is usually time-consuming and expensive, we have $N \gg K$ in practice. The complete retraining process is shown in Alg. 1 (App. A).

Given the training dataset $\mathcal{S}_t$ and the post-best-response distribution $P_{XY}^t$, we can define their associated ***qualification rates*** as the proportion/probability of agents that are qualified, i.e.,

$$Q(\mathcal{S}_t) = \mathbb{E}_{(x,y) \in \mathcal{S}_t} [y] ; \qquad Q(P^t) = \mathbb{E}_{(x,y) \sim P_{XY}^t} [y] ,$$

For the classifier $f_t$ deployed on marginal feature distribution $P_X^t$, we define ***acceptance rate*** as the probability that agents are classified as positive, i.e.,

$$A(f_t, P^t) = \mathbb{E}_{x \sim P_X^t} [f_t(x)].$$

Since $\mathcal{S}_t$ is related to random sampling at all $t$, the resulting classifier $f_t$ and agents' best responses are also random. Denote $D_{XY}^t$ as the probability distribution of sampling from $\mathcal{S}_t$ and recall that $D_{XY}^o$ is the distribution for *human-annotated* $\mathcal{S}_{o,t}$, we can further define the expectations of qualification/acceptance rate $Q(\mathcal{S}_t), Q(P^t), A(f_t, P^t)$ over the training dataset:

$$\overline{q}_t := \mathbb{E}_{\mathcal{S}_t}[Q(\mathcal{S}_t)]; \qquad q_t := \mathbb{E}_{\mathcal{S}_{t-1}}[Q(P^t)]; \qquad a_t := \mathbb{E}_{\mathcal{S}_t} \left[ A(f_t, P^t) \right]$$

where $\overline{q}_t$ is the expected qualification rate of agents in the training set; $q_t$ is the expected actual qualification rate of agents after they best respond, note that the expectation is taken with respect to $\mathcal{S}_{t-1}$ because the distribution $P_{XY}^t$ is the result of agents best responding to $f_{t-1}$ which is trained with $\mathcal{S}_{t-1}$; $a_t$ is the expected acceptance rate of agents at time $t$.

**Dynamics of qualification rate & acceptance rate.** Under the model retraining process, both the model $f_t$ and agents' distribution $P_{XY}^t$ change over time. One goal of this paper is to understand how the agents and the ML model interact and impact each other in the long run. Specifically, we are interested in the dynamics of the following variables:

1. **Qualification rate** $q_t$: it measures the qualification of agents and indicates the *social welfare*.
2. **Acceptance rate** $a_t$: it measures the likelihood that an agent can receive positive outcomes and indicates the *applicant welfare*.
3. **Classifier bias** $\Delta_t = |a_t - q_t|$: it is the discrepancy between the acceptance rate and the true qualification rate, measuring how well the decision-maker can approximate agents' actual qualification rate and can be interpreted as *decision-maker welfare*.

While it is difficult to derive the dynamics of $a_t$ and $q_t$ explicitly, we can first work out the dynamics of $\overline{q}_t$ using the law of total probability (details in App. G.1), i.e.,

$$\overline{q}_t = \frac{tN+(t-1)K}{(t+1)N+tK} \cdot \overline{q}_{t-1} + \frac{N}{(t+1)N+tK} \cdot a_{t-1} + \frac{K}{(t+1)N+tK} \cdot \overline{q}_0 \tag{3}$$

Then, we explore relations between $\overline{q}_t$ and $a_t$ (or $q_t$). By leveraging such relations and equation 3, we can further study the dynamics of $a_t$ (or $q_t$).

**Objectives.** This paper studies the above dynamics and we aim to answer the following questions: 1) How do the qualification rate $q_t$, acceptance rate $a_t$, and classifier bias $\Delta_t$ evolve under the dynamics? 2) How can the evolution of the system be affected by the decision-maker's retraining process? 3) What are the impacts of the decision-maker's systematic bias? 4) If we further consider agents from multiple social groups, how can the retraining process affect inter-group fairness?

## 3 Dynamics of the Agents and Model

In this section, we examine the evolution of qualification rate $q_t$, acceptance rate $a_t$, and classifier bias $\Delta_t$. We aim to understand how *applicant welfare* (Sec. 3.1), *social welfare* (Sec. 3.2), and *decision-maker welfare* (Sec. 3.3) are affected by the retraining process in the long run.

Because the acceptance rate $a_t := \mathbb{E}_{\mathcal{S}_t} \left[ A(f_t, P^t) \right]$ and qualification rate $q_t := \mathbb{E}_{\mathcal{S}_{t-1}}[Q(P^t)]$ depend on agent post-best-response distribution $P_{XY}^t$ and the classifiers, we can indeed identify all sources that affect the evolution of these quantifies (details are in App. G.2):

- $\overline{q}_t$: *Expected qualification rate of agents in training set.*
- $\delta(D^t, \mathcal{F})$ : *Algorithmic bias* that measures how well hypothesis class $\mathcal{F}$ can approximate the training data distribution $D^t_{Y|X}$. It universally exists in the *PAC learning* framework.
- $\delta^t_{BR}$: *Strategic shift* caused by agents' best responses to $f_{t-1}$ at $t$.

Note that $\delta^t_{BR}$ and $\overline{q}_t$ closely affect each other and cannot be decoupled quantitatively. Meanwhile, $\delta(D^t, \mathcal{F})$ only depends on how well the hypothesis class $\mathcal{F}$ fits $P^t_{Y|X}$. Since $\mathcal{F}$ is pre-determined, we ignore $\delta(D^t, \mathcal{F})$ in our theoretical analysis (Assumption 3.1). However, all experiments in Sec. 5 and Appendix naturally capture $\delta(D^t, \mathcal{F})$ and the results are consistent with theorems.

**Assumption 3.1.** Under the retraining process, algorithmic bias $\delta(D^t, \mathcal{F})$ is negligible.

Finally, we further assume the monotone likelihood ratio property holds for $D^o_{XY}$ and $P_{XY}$.

**Assumption 3.2.** Let $x[m]$ be $m$-dimension of $x \in \mathbb{R}^d$, then $D^o_{Y|X}(1|x)$ and $P_{Y|X}(1|x)$ is continuous and monotonically increasing in $x[m]$, $\forall m = 1, \cdots, d$.

Note that the Assumption 3.2 is mild and widely used in previous literature (Zhang et al., 2022). It can be satisfied by many distributional families such as exponential, Gaussian, and mixtures of exponential/Gaussian. It implies that agents are more likely to be qualified as feature value increases.

### 3.1 APPLICANT WELFARE: THE DYNAMICS OF ACCEPTANCE RATE

We first examine the dynamics of $a_t = \mathbb{E}_{\mathcal{S}_t}[A(f_t, P^t)]$. Intuitively, when $\delta(D^t, \mathcal{F})$ is negligible (Assumption 3.1), all classifiers can fit the training data well. Then the model-annotated samples $\mathcal{S}_{m,t-1}$ generated from post-best-response agents would have a higher qualification rate than the qualification rate of $\mathcal{S}_{t-1}$ (i.e., $\overline{q}_{t-1}$). As a result, the training data $\mathcal{S}_t$ augmented with $\mathcal{S}_{m,t-1}$ has a higher proportion of qualified agents $\overline{q}_t$ than $\overline{q}_{t-1}$, thereby producing a more "generous" classifier $f_t$ with a larger $a_t$. This reinforcing process can be formally stated in Thm. 3.3.

**Theorem 3.3** (Evolution of $a_t$). *Under the retraining process, the acceptance rate of the agents that join the system increases over time, i.e., $a_t > a_{t-1}$, $\forall t \geqslant 1$.*

We prove Thm. 3.3 by mathematical induction in App. G.3. Fig. 3 below illustrates Thm. 3.3 by showing how agents' best responses can reshape training data $\mathcal{S}_t$ and classifier $f_t$. When agents best respond, the decision-maker tends to accept more and more agents. Indeed, we can further show that when the number of *model-annotated* samples $N$ is sufficiently large compared to the number of *human-annotated* samples $K$, the classifier will accept all agents in the long run (Prop. 3.4).

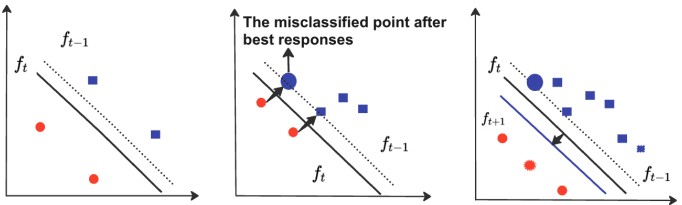

Figure 3: Illustration of increased acceptance rate $a_t$. The left plot shows the training dataset $\mathcal{S}_t$ contains 2 unqualified (red circle) and 2 qualified agents (blue square) and $a_t$ is 0.5. The middle plot shows the new agents coming at $t$ best respond to $f_{t-1}$. After the best responses, 3 of 4 agents are qualified (blue square) and 1 is still unqualified (blue circle). However, all 4 agents are annotated as "qualified" (blue). The right plot shows the training dataset $\mathcal{S}_{t+1}$ contains all points of the left and middle plot, plus two new human-annotated points (points with dashed edges). All blue points are labeled as 1 and the red points are labeled as 0. So the qualification rate $\overline{q}_{t+1}$ of $\mathcal{S}_{t+1}$ becomes larger and $f_{t+1}$ accepts a higher proportion of agents ($a_{t+1}$ is 0.58).

**Proposition 3.4.** *For any set of $P_{XY}, D^o, B$, there exists a threshold $\lambda > 0$ such that $\lim_{t \to \infty} a_t = 1$ whenever $\frac{K}{N} < \lambda$.*

The specific value of $\lambda$ in Prop. 3.4 depends on $P_{XY}, D^o, B$, which is difficult to find analytically. Nonetheless, we can demonstrate in Sec. 5 that when $\frac{K}{N} = 0.05$, $a_t$ tends to approach 1 in various datasets. Since the *human-annotated* samples are often difficult to attain (due to time and labeling costs), the condition in Prop. 3.4 is easy to satisfy in practice.

## 3.2 Social welfare: the dynamics of qualification rate

Next, we study the dynamics of qualification rate $q_t = \mathbb{E}_{\mathcal{S}_{t-1}}[Q(P^t)]$. Unlike $a_t$ which always increases during the retraining process, the evolution of $q_t$ is more complicated and depends on agents' prior-best-response distribution $P_{XY}$.

Specifically, let $q_0 = Q(P) = \mathbb{E}_{(x,y)\sim P_{XY}}[y]$ be the initial qualification rate, then the difference between $q_t$ and $q_0$ can be interpreted as the amount of *improvement* (i.e., increase in label) agents gain from their best responses at $t$. This is determined by (i) the proportion of agents that decide to change their features at costs (depends on $P_X$), and (ii) the improvement agents can expect upon changing features (depends on $P_{Y|X}$). Thus, the dynamics of $q_t$ depend on $P_{XY}$. Despite the intricate nature of dynamics, we can still derive a sufficient condition under which $q_t$ decreases monotonically.

**Theorem 3.5** (Evolution of $q_t$). *Let $F_X(x)$ be the cumulative density function corresponding to $P_X$. Denote $\mathcal{J} = \{x|f_0(x) = 0\}$ as the half-space in $\mathbb{R}^d$ determined by the classifier $f_0$ trained with $\mathcal{S}_{o,0}$. Under the retraining process, if $F_X$ and $P_{Y|X}(1|x)$ are convex on $\mathcal{J}$, then $q_{t+1} < q_t$, $\forall t \geqslant 1$.*

Note that $q_{t+1} < q_t$ in Thm. 3.5 holds only for $t \geqslant 1$. Because agents can only improve their labels from their best responses, prior-best-response $q_0$ always serves as the lower bound of $q_t$. The half-space $\mathcal{J}$ in Thm. 3.5 specifies the region in feature space where agents have incentives to change their features. The convexity of $F_X$ and $P_{Y|X}(1|x)$ ensure that as $f_t$ evolves from $t = 1$: (i) fewer agents choose to improve their features, and (ii) agents expect less improvement from feature changes. Thus, $q_t$ decreases over time. The proof and a more general analysis are shown in App. G.5.

Indeed, the condition in Thm. 3.5 can be satisfied by common distributions $P_X$ (e.g., Uniform, Beta when $\alpha > \beta$) and labeling functions $P_{Y|X}(1|x)$ (e.g., linear function, quadratic functions with degree greater than 1). Other distributions (e.g., Gaussian) and labeling functions (e.g., logistic function) can also satisfy the condition if $F_X$ and $P_{Y|X}(1|x)$ are convex on $x \in \mathcal{J}$. We also show that Thm. 3.5 is valid under diverse experimental settings (Sec. 5, App. E, App. F).

## 3.3 Decision-maker welfare: the dynamics of classifier bias

Sec. 3.1 and 3.2 show that as the classifier $f_t$ gets updated over time, agents are more likely to get accepted ($a_t$ increases). However, their true qualification rate $q_t$ (after the best response) may actually decrease. It indicates that the decision-maker's misperception about agents varies over time. Thus, this section studies the dynamics of classifier bias $\Delta_t = |a_t - q_t|$. Our results show that the evolution of $\Delta_t$ is largely affected by the systematic bias and its magnitude $\mu(D^o, P)$ (Def. 2.1).

**Theorem 3.6** (Evolution of $\Delta_t$). *Under the retraining process and the conditions in Thm. 3.5:*

1. *If systematic bias does not exist (i.e., $\mu(D^o, P) = 0$), then $\Delta_t$ increases over time.*

2. *If the decision-maker overestimates agent qualification ($\mu(D^o, P) > 0$), then $\Delta_t$ increases.*

3. *If the decision-maker underestimates agent qualification ( $\mu(D^o, P) < 0$), then $\Delta_t$ **either** monotonically decreases **or** first decreases but then increases.*

Thm. 3.6 highlights the potential risks of the model retraining process and is proved in App. G.6. Originally, the purpose of retraining the classifier was to ensure accurate decisions on the targeted population. However, in the presence of strategic agents, the retraining may lead to adverse outcomes by amplifying the classifier bias. Meanwhile, though systematic bias is usually an undesirable factor to eliminate when learning ML models, it may help mitigate classifier bias to improve the *decision-maker welfare* in the retraining process, i.e., $\Delta_t$ decreases when $\mu(D^o, P) < 0$.

## 3.4 Intervention to stabilize the dynamics

Sec. 3.1- 3.3 show that as the model is retrained from strategic agents, $a_t, q_t, \Delta_t$ are unstable and may change monotonically over time. Next, we introduce an effective approach to stabilize the system.

From the above analysis, we know that one reason that makes $q_t, a_t, \Delta_t$ evolve is agent's best response, i.e., agents improve their features strategically to be accepted by the most recent model, which leads to a higher qualification rate of *model-annotated* samples (and the resulting training data), eventually causing $a_t$ to deviate from $q_t$. Thus, to mitigate such deviation, we can improve the

quality of model annotation. Our method is proposed based on this idea, which uses a *probabilistic sampler* (Taori & Hashimoto, 2023) when producing *model-annotated* samples.

Specifically, at each time $t$, instead of adding $\mathcal{S}_{m-1,o} = \{x_{t-1}^i, f_{t-1}(x_{t-1}^i)\}_{i=1}^N$ (samples annotated by $f_{t-1}$) to training data $\mathcal{S}_t$ (equation 2), we use a probabilistic model $\Phi_{t-1} : \mathbb{R}^d \to [0,1]$ to annotate each sample according to the following: *for each sample $x$, we label it as $1$ with probability $\Phi_{t-1}(x)$, and as $0$ otherwise.* Here $\Phi_{t-1}(x) \approx D_{Y|X}^{t-1}(1|x)$ is the estimated posterior probability learned from $\mathcal{S}_{t-1}$ (e.g., logistic model). We call the procedure *refined retraining process* if model-annotated samples are generated in this way based on a probabilistic sampler.

Fig. 3 also illustrates the idea: after agents best respond to $f_{t-1}$ (middle plot), their features improve and $f_t$ will label both as $1$. By contrast, a probabilistic sampler $\Phi_t$ only labels a fraction of them as $1$ to produce a smaller $\bar{q}_{t+1}$. This alleviates the influence of agents' best responses to stabilize the dynamics of $a_t, q_t, \Delta_t$. In App. F.3, we also compare the evolution of $a_t, q_t, \Delta_t$ under *refined retraining process* with that under the original retraining process, and the results validate our approach.

## 4 LONG-TERM FAIRNESS DYNAMICS UNDER STRATEGIC RETRAINING

**Dynamics without fairness interventions.** In this section, we focus on the long-term fairness dynamics of the retraining process. We first state that the decision-maker easily has systematic bias (Adebayo et al., 2022; Bareinboim & Pearl, 2012; Alvero et al., 2020) (see App. B for motivating examples). In this section, we consider scenarios where agents come from two groups $i, j$ with different sensitive attributes, and the decision-maker, uses group-dependent classifiers to make decisions about two groups of agents. We assume the initial qualification $q_0^i \geqslant q_0^j$ and both groups have the same cost matrix $B$ to change features. Denote the systematic bias to $i, j$ as $\mu_i, \mu_j$ where $\mu_i \geqslant \mu_j$. This is reasonable because the group with lower qualifications is usually under-represented. To measure the unfairness, we consider the metric *demographic parity* (DP) (Feldman et al., 2015), which measures the unfairness between two groups $i, j$ as the discrepancy in their acceptance rates $|a_t^i - a_t^j|$. The extension to other commonly used fairness metrics is discussed in App. D.2.

First, if applying the original retraining process for both groups, then the decision-maker is expected to admit **all** agents in the long run and it ultimately preserves fairness, but the **classifier bias** will be maximized. However, the dynamics in the middle of the retraining cannot be determined without knowing the feature distributions of both groups. By contrast, applying *refined retraining process* on both groups can stabilize the dynamics, but it cannot mitigate the systematic bias of the decision-maker. In the left plot of Fig. 4, we see that under the *refined retraining process*, the unfairness between groups is always around

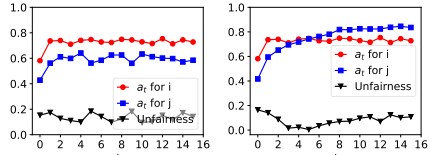

Figure 4: Comparison of unfairness (DP) when *refined retraining process* is applied to both groups (left) and when it is only applied to group $i$ (right) under dataset 2.

$0.2$. Instead, if the decision-maker only applies the *refined retraining process* on $i$ while keeping the original retraining process on $j$, then perfect fairness will be achieved in the middle of the retraining process, but the model becomes unfair again as the retraining goes on.

**Theorem 4.1** (Promote fairness through the *early stopping* mechanism). *When $q_0^i \geqslant q_0^j$ and $\mu_i \geqslant \mu_j$, if the decision-maker applies the refined retraining process to group $i$ while applying the original retraining process to group $j$, then $|a_t^i - a_t^j|$ will first decrease to be close to $0$, and then increase.*

Thm. 4.1 implies that the decision-maker can monitor the unfairness at each round, and executes the *early stopping* mechanism to attain almost perfect DP fairness. As shown in the right plot of Fig. 4, the unfairness is minimized at $t = 5$ under the proposed method.

**Fairness interventions at each round.** Since both the original retraining process and the *refined retraining process* are unable to maintain demographic parity among groups. We consider fairness interventions at each round to ensure the deployment of fair models in App. D.2. Specifically, we examine the dynamics of the qualification rate and acceptance rate for both groups under fairness interventions. The results show that fairness interventions under the original retraining process still cause the qualification rate and acceptance rate to change monotonically, but the intervention on the *refined retraining process* can produce stable and fair classifiers.

## 5 EXPERIMENTS

We conduct experiments on two synthetic datasets (Uniform, Gaussian), one semi-synthetic dataset (German Credit (Hofmann, 1994)), and one real dataset (Credit Approval (Quinlan, 2017)) to validate the theorems and proposed methods. Note that only the Uniform dataset satisfies all assumptions and the conditions in the above theoretical analysis, while the Gaussian dataset and German Credit dataset violate the conditions in Thm. 3.5. The Credit Approval dataset violates all assumptions and conditions of the main paper. The decision-maker trains logistic regression models for all experiments using stochastic gradient descent (SGD) over $T$ steps. We present the experimental results of the Gaussian and German Credit datasets in this section, while the results for Uniform and Credit Approval data are similar and shown in App. E.

**Gaussian data.** We consider a synthetic dataset with Gaussian distributed $P_X$. $P_{Y|X}$ is logistic and satisfies Assumption 3.2 but not the conditions of Thm. 3.5. We assume agents have two independent features $X_1, X_2$ and

Table 1: Gaussian Dataset Setting

| $P_{Xk}(x_k)$ | $P_{Y|X}(1|x)$ | $n, r, T, q_0$ |
|---|---|---|
| $\mathcal{N}(0, 0.5^2)$ | $(1 + \exp(x_1 + x_2))^{-1}$ | $100, 0.05, 15, 0.5$ |

are from two groups $i, j$ with different sensitive attributes but identical joint distribution $P_{XY}$. Their cost matrix is $B = \begin{bmatrix} 5 & 0 \\ 0 & 5 \end{bmatrix}$ and the initial qualification rate is $q_0 = 0.5$. We assume the decision-maker has a systematic bias by overestimating (resp. underestimating) the qualification of agents in the advantaged group $i$ (resp. disadvantaged group $j$), which is modeled as increasing $D^o_{Y|X}(1|x)$ to be 0.1 larger (resp. smaller) than $P_{Y|X}(1|x)$ for group $i$ (resp. group $j$). For the retraining process, we let $r = \frac{K}{N} = 0.05$ (i.e., the number of model-annotated samples $N = 2000$, which is sufficiently large compared to the number of human-annotated samples $K = 100$). Table 1 summarizes the dataset information, and the joint distributions are visualized in App. F.1.

We first verify the results in Sec. 3 by illustrating the dynamics of $a_t, q_t, \Delta_t$ for both groups (Fig. 5a). Since our analysis neglects the algorithmic bias and the evolution results are in expectation, we perform $n = 100$ independent runs of experiments for every parameter configuration and show the averaged outcomes. The results are consistent with Thm. 3.3, 3.5 and 3.6: (i) acceptance rate $a_t$ (red curves) increases monotonically; (ii) qualification rate $q_t$ decreases monotonically starting from $t = 1$ (since strategic agents only best respond from $t = 1$); (iii) classifier bias $\Delta_t$ evolves differently for different groups and it may reach the minimum after a few rounds of retraining. Next, we verify whether the *early stopping* mechanism of the retraining process proposed in Thm. 4.1 can promote fairness. Fig. 5b shows that the decision-maker attains almost *perfect* fairness at $t = 5$. However, as discussed in Sec. 4, although fairness can be enhanced, it only ensures both groups have a similar classifier bias $\Delta_t$ but cannot reduce such bias.

Besides, while we assume agents at round $t$ have perfect knowledge of the classifier $f_{t-1}$, Jagadeesan et al. (2021) pointed out agents may have a noisy knowledge in practice. To test the robustness of our theoretical results against agent noisy response, we assume agents estimate their classification result as $\hat{f}_t(x) = f_t(x) + \epsilon$ where $\epsilon \sim N(0, 0.1)$. We present the dynamics of $a_t, q_t, \Delta_t$ for both groups in Fig. 7a which are quite similar to Fig. 5a, demonstrating the robustness of our theorems.

**German Credit dataset (Hofmann, 1994).** This dataset includes features for predicting individuals' credit risks. It has 1000 samples and 19 numeric features, which are used to construct a larger-scaled dataset. Specifically, we fit a kernel density estimator for all 19 features to generate 19-dimensional features, the corresponding labels are sampled from the distribution $P_{Y|X}$ which is estimated from data by fitting a logistic classifier with 19 features. Given this dataset, the first 10 features are used to train the classifiers. The attribute "sex" is regarded as the sensitive attribute. The systematic bias is created by increasing/decreasing $P_{Y|X}$ by 0.05. Other parameters $n, r, T, q_0$ are the same as Table 1. Since $P_{Y|X}$ is a logistic function, Assumption 3.2 can be satisfied easily as illustrated in App. F.1.

We first verify the results in Sec. 3 by illustrating the dynamics of $a_t, q_t, \Delta_t$ for both groups. The results are shown in Fig. 6a and are consistent with Thm. 3.3, 3.5 and 3.6: (i) acceptance rate $a_t$ (red curves) always increases; (ii) qualification rate $q_t$ (blue curves) decreases starting from $t = 1$ (since strategic agents only best respond from $t = 1$); (iii) classifier bias $\Delta_t$ (black curves) evolve differently for different groups. In the right plot of Fig. 6a, $\Delta^j_t$ reaches the minimum at $t = 2$, suggesting the best time for the decision-maker to stop retraining to maximize its welfare. We also evaluate the *early stopping* mechanism of the retraining and verify Thm. 4.1. Fig. 6b shows the unfairness decreases

first and is minimized at $t = 9$. Finally, similar to Fig. 7a, Fig. 7b demonstrates the results are still robust under the noisy setting.

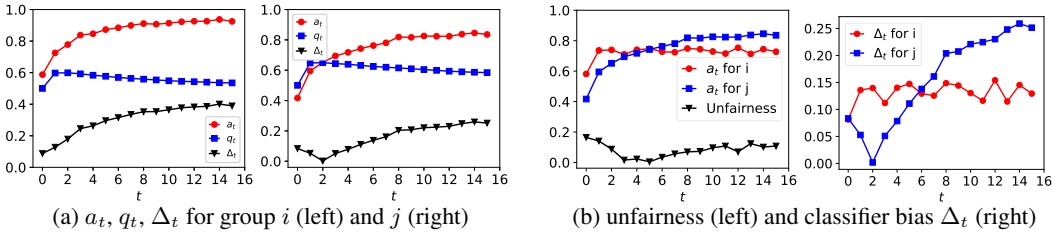

(a) $a_t, q_t, \Delta_t$ for group $i$ (left) and $j$ (right)

(b) unfairness (left) and classifier bias $\Delta_t$ (right)

Figure 5: Dynamics of $a_t, q_t, \Delta_t$ and unfairness $|a_t^i - a_t^j|$ on Gaussian dataset.

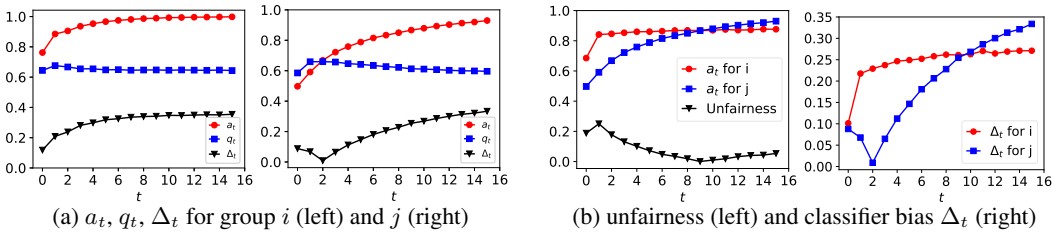

(a) $a_t, q_t, \Delta_t$ for group $i$ (left) and $j$ (right)

(b) unfairness (left) and classifier bias $\Delta_t$ (right)

Figure 6: Dynamics of $a_t, q_t, \Delta_t$ and unfairness $|a_t^i - a_t^j|$ on German Credit dataset.

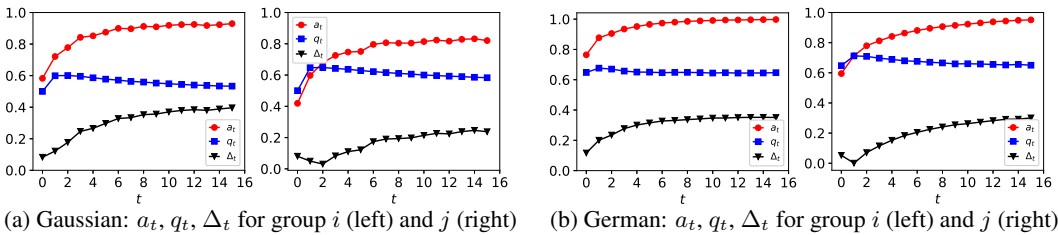

(a) Gaussian: $a_t, q_t, \Delta_t$ for group $i$ (left) and $j$ (right)

(b) German: $a_t, q_t, \Delta_t$ for group $i$ (left) and $j$ (right)

Figure 7: Dynamics of $a_t, q_t, \Delta_t$ under the noisy setting.

**More comprehensive experiments in App. F.** App. F.1 describes experimental setups. App. F.2 demonstrates the additional results to verify Thm. 3.3 and Thm. 3.6 under different $r = \frac{K}{N}$, where we observe the same trends under different $r$, but $a_t, q_t$ change with different rates. It also provides results on how $a_t, q_t, \Delta_t$ change under the following situations: (i) the longer-term dynamics when $T$ is very large; (ii) **no** systematic bias; (iii) all training examples are *human-annotated* or *model-annotated*; (iv) agents have different costs of changing different features. App. F.3 presents more dynamics under *refined retraining process* to illustrate how it stabilizes the retraining; App. F.4 illustrates the evolution of unfairness under various $r$; App. F.5 presents more experiments under the noisy setting, while App. F.6 compares the situations when agents are non-strategic with the ones when they are strategic, revealing that the strategic feedback of agents causes $a_t, q_t$ to diverge.

## 6 CONCLUSION & LIMITATIONS

This paper studies the dynamics where strategic agents interact with an ML system retrained over time with *model-annotated* and *human-annotated* samples. We rigorously studied the evolution of *applicant welfare*, *decision-maker welfare*, and *social welfare*. Such results highlight the potential risks of retraining classifiers when agents are strategic. The paper also proposed solutions to stabilizing dynamics and improving fairness. However, our theoretical results rely on certain assumptions and we should first verify these conditions before adopting the results of this paper, which may be challenging in real-world applications. Finally, though *early stopping* is a simple yet powerful mechanism to promote fairness, it remains an interesting problem to ensure fairness during endless retraining.

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

## A  THE RETRAINING PROCESS

---

**Algorithm 1** retraining process

---

**Require:** Joint distribution $D_{XY}^o$ for any $\mathcal{S}_{o,t}$, Hypothesis class $\mathcal{F}$, the number of the initial training samples and agents coming per round $N$, the number of decision-maker-labeled samples per round $K$.

**Ensure:** Model deployments over time $f_0, f_1, f_2, \ldots$

1: $S_0 = S_{o,0} \sim D_{XY}^o = \{x_0^i\}_{i=1}^N$
2: At $t = 0$, deploy $f_0 \sim \mathcal{F}(\mathcal{S}_0)$
3: **for** $t \in \{1, \ldots \infty\}$ **do**
4:     $N$ agents gain knowledge of $f_{t-1}$ and best respond to it, resulting in $\{x_t^i\}_{i=1}^N$
5:     $\mathcal{S}_{m,t} = \{x_{t-1}^i, f_{t-1}(x_{t-1}^i)\}_{i=1}^N$ consists of the model-labeled samples from round $t-1$.
6:     $\mathcal{S}_{o,t} \sim D_{XY}^o$ consists of the new $K$ decision-maker-labeled samples.
7:     $\mathcal{S}_t = \mathcal{S}_{t-1} \cup \mathcal{S}_{m,t} \cup \mathcal{S}_{o,t}$
8:     Deploy $f_t \sim \mathcal{F}(\mathcal{S}_t)$ on the incoming $N$ agents who best respond to $f_{t-1}$ with the resulting joint distribution $P_{XY}^t$.
9: **end for**

---

## B  MOTIVATING EXAMPLES OF THE SYSTEMATIC BIAS

Def. 2.1 highlights the systematic nature of the decision-maker's bias. This bias is quite ubiquitous when labeling is not a trivial task. It almost always the case when the decision-maker needs to make human-related decisions. We provide the following motivating examples of systematic bias with supporting literature in social science:

1. *College admissions*: consider experts in the admission committee of a college that obtains a set of student data and wants to label all students as "qualified" or "unqualified". The labeling task is much more complex and subjective than the ones in computer vision/natural language processing which have some "correct" answers. Therefore, the experts in the committee are prone to bring their "biases" towards a specific population sharing the same sensitive attribute into the labeling process including:

   (a) *Implicit bias:* the experts may have an implicit bias they are unaware of to favor/discriminate against students from certain groups. For instance, a famous study (Capers IV et al., 2017) reveals admission committee members at the medical school of the Ohio State University unconsciously have a "better impression" towards white students; Alvero et al. (2020) finds out that even when members in an admission committee do not access the sensitive attributes of students, they unconsciously infer them and discriminate against students from the minority group.

   (b) *Selection bias:* the experts may have insufficient knowledge of the under-represented population due to the selection bias (Bareinboim & Pearl, 2012) because only a small portion of them were admitted before. Thus, experts may expect a lower qualification rate from this population, resulting in more conservative labeling practices. The historical stereotypes created by selection bias are difficult to erase.

2. *Loan applications*: consider experts in a big bank that obtains data samples from some potential applicants and wants to label them as "qualified" or "unqualified". Similarly, the experts are likely to have systematic bias including:

   (a) *Implicit bias:* similarly, Brock & De Haas (2023) conduct a lab-in-the-field experiment with over 300 Turkish loan officers to show that they bias against female applicants even if they have identical profiles as male applicants.

   (b) *Selection bias:* when fewer female applicants are approved historically, the experts have less knowledge on females (i.e., whether they will actually default or repay), thereby tending to stay conservative.

## C   RELATED WORK

### C.1   STRATEGIC CLASSIFICATION

**Strategic classification without label changes.** Our work is mainly based on an extensive line of literature on strategic classification (Hardt et al., 2016; Ben-Porat & Tennenholtz, 2017; Dong et al., 2018; Jagadeesan et al., 2021; Levanon & Rosenfeld, 2022; Braverman & Garg, 2020; Izzo et al., 2021; Chen et al., 2020b; Ahmadi et al., 2021; Tang et al., 2021; Zhang et al., 2020; 2022; Eilat et al., 2022; Liu et al., 2022; Lechner & Urner, 2022; Horowitz & Rosenfeld, 2023). These works assume the agents are able to best respond to the policies of the decision-maker to maximize their utilities. Most works modeled the strategic interactions between agents and the decision-maker as a repeated Stackelberg game where the decision-maker leads by publishing a classifier and the agents immediately best respond to it. The earliest line of works focused on the performance of regular linear classifiers when strategic behaviors never incur label changes (Hardt et al., 2016; Ben-Porat & Tennenholtz, 2017; Dong et al., 2018; Chen et al., 2020b), while the later literature added noise to the agents' best responses (Jagadeesan et al., 2021), randomized the classifiers (Braverman & Garg, 2020) and limited the knowledge of the decision-maker (Tang et al., 2021). Levanon & Rosenfeld (2022) proposed a generalized framework for strategic classification and a *strategic hinge loss* to better train strategic classifiers, but the strategic behaviors are still not assumed to cause label changes.

**Strategic classification with label changes.** Several other lines of literature enable strategic behaviors to cause label changes. The first line of literature mainly focuses on incentivizing improvement actions where agents have budgets to invest in different actions and only some of them cause the label change (improvement) (Kleinberg & Raghavan, 2020; Harris et al., 2021; Bechavod et al., 2022; Jin et al., 2022; Chen et al., 2020a; Haghtalab et al., 2020; Alon et al., 2020; Bechavod et al., 2021; Raab & Liu, 2021). The other line of literature focuses on *causal strategic learning* (Miller et al., 2020; Shavit et al., 2020; Horowitz & Rosenfeld, 2023; Harris et al., 2022; Yan et al., 2023). These works argue that every strategic learning problem has a non-trivial causal structure which can be explained by a *structural causal model*, where intervening on causal nodes causes improvement and intervening on non-causal nodes means manipulation.

**Performative prediction.** Several works consider *performative prediction* as a more general setting where the feature distribution of agents is a function of the classifier parameters. Perdomo et al. (2020) first formulated the prediction problem and provided iterative algorithms to find the stable points of the model parameters. Izzo et al. (2021) modified the gradient-based methods and proposed the `Perfgrad` algorithm. Hardt et al. (2022) elaborated the model by proposing *performative power*.

**Retraining under strategic settings**

Most works on learning algorithms under strategic settings consider developing robust algorithms that the decision-maker only trains the classifier once (Hardt et al., 2016; Tang et al., 2021; Levanon & Rosenfeld, 2022; Jagadeesan et al., 2021), while *Performative prediction*(Perdomo et al., 2020) focuses on developing online learning algorithms for strategic agents. There are only a few works (Horowitz & Rosenfeld, 2023; Rosenfeld et al., 2020) which permit retraining the Strategic classification models. However, all these algorithms assume that the decision-maker has access to a new training dataset containing both agents' features and labels at each round. There is no work considering the dynamics under the retraining process with *model-annotated* samples.

### C.2   BIAS AMPLIFICATION DURING RETRAINING

There has been an extensive line of study on the computer vision field about how machine learning models amplify the dataset bias, while most works only focus on the one-shot setting where the machine learning model itself amplifies the bias between different groups in one training/testing round (Hall et al., 2022; Adebayo et al., 2022). In recent years, another line of research focuses on the amplification of dataset bias under *model-annotated data* where ML models label new samples on their own and add them back to retrain themselves(Leino et al., 2018; Dinan et al., 2019; Wang et al., 2019; Zhao et al., 2017; Adam et al., 2022; Sculley et al., 2015; Ensign et al., 2018; Mansoury et al., 2020; Adam et al., 2020). These works study the bias amplification in different practical fields including resource allocation (Ensign et al., 2018), computer vision (Wang et al., 2019), natural language processing (Zhao et al., 2017) and clinical trials (Adam et al., 2020). The most related work is (Taori & Hashimoto, 2023) which studied the influence of retraining in the non-strategic setting.

Also, there is a work (Adam et al., 2022) touching on the data feedback loop under performative setting, but it focused on empirical experiments under medical settings where the feature distribution shifts are mainly caused by treatment and the true labels in historical data are highly accessible. Besides, the data feedback loop is also related to recommendation systems. where extensive works have studied how the system can shape users' preferences and disengage the minority population (Schmit & Riquelme, 2018; Sinha et al., 2016; Jiang et al., 2019; Mansoury et al., 2020). However, previous literature did not touch on the retraining process.

### C.3 MACHINE LEARNING FAIRNESS IN STRATEGIC CLASSIFICATION

Several works have considered how different fairness metrics (Feldman et al., 2015; Hardt et al., 2016; Gupta et al., 2019; Guldogan et al., 2022) are influenced in strategic classification (Liu et al., 2019; Zhang et al., 2020; Liu et al., 2020; Zhang et al., 2022). The most related works (Liu et al., 2020; Zhang et al., 2022) studied how strategic behaviors and the decision-maker's awareness can shape long-term fairness. They deviated from our paper since they never considered retraining and the strategic behaviors never incurred label changes.

## D ADDITIONAL DISCUSSIONS

### D.1 HUMAN-ANNOTATED SAMPLES DRAWN FROM $P_X^t$

In this section, we consider the situation where all *human-annotated* samples at $t$ are drawn from the *post-best-response* distribution $P_X^t$. This will change equation 3 to the following:

$$\overline{q'}_t = \frac{tN+(t-1)K}{(t+1)N+tK} \cdot \overline{q'}_{t-1} + \frac{N}{(t+1)N+tK} \cdot a'_{t-1} + \frac{K}{(t+1)N+tK} \cdot q^*_{t-1} \tag{4}$$

where $q'$ and $a'$ denote the new qualification rate and acceptance rate, and $q^*_{t-1}$ stands for the qualification rate of the human annotations on features drawn from $P_X^{t-1}$. Note that the only difference lies in the third term of the RHS which changes from $\overline{q}_0$ to $q^*_{t-1}$. Our first observation is that $q^*_{t-1}$ is never smaller than $\overline{q}_0$ because the best response will not harm agents' qualifications. With this observation, we can derive Prop. D.1.

**Proposition D.1.** $a'_t \geqslant a_t$ *holds for any* $t \geqslant 1$. *If Prop. 3.4 further holds, we also have* $a'_t \to 1$.

Prop. D.1 can be proved easily by applying the observation stated above. However, note that unlike $a_t$, $a'_t$ is not necessarily monotonically increasing.

### D.2 ADDITIONAL DISCUSSIONS ON FAIRNESS

**Other fairness metrics.** It is difficult to derive concise results considering other fairness metrics including *equal opportunity* and *equal improvability* because the data distributions play a role in determining these metrics. However, Theorem 3.5 states the true qualification rate of the agent population is likely to decrease, suggesting the retraining process may do harm to improvability. Meanwhile, when the acceptance rate $a_t$ increases for the disadvantageous group, the acceptance rate of the qualified individuals will be likely to be better, but it is not guaranteed because the feature distribution of the qualified individuals also changes because we assume the strategic behaviors are causal which may incur label changes.

**Fairness intervention at each round with original retraining process.** In Sec. 4, we show that even if the retraining process promotes intergroup fairness, fairness is likely to be achieved only when the decision-maker overestimates the qualification of both groups. In this section, we consider another approach by applying a "hard fairness intervention" on both groups after retraining the classifier at each round. Specifically, let $f_t^i = \mathbf{1}(h_t^i(x) \geqslant \theta_t^i)$, $f_t^j = \mathbf{1}(h_t^j(x) \geqslant \theta_t^j)$ be the original optimal classifiers for two groups without fairness constraints. Here $h$ is used to predict the conditional probability of an agent having label 1 given their feature $x$. Note that $f_t^i, f_t^j$ are trained from $\mathcal{S}_t^i, \mathcal{S}_t^j$ which do not include agents coming at $t$. Thus, to ensure DP fairness on the agents coming at $t$ with joint distributions $P_{XY}^{it}, P_{XY}^{jt}$, the decision-maker needs to post-process $f_t^i, f_t^j$ to further satisfy fairness constraint. Specifically, let the fair-optimal models for two groups be $\widetilde{f}_t^i, \widetilde{f}_t^j$, the decision-maker obtains them by adjusting the thresholds $\theta_t^i, \theta_t^j$ to $\widetilde{\theta}_t^i, \widetilde{\theta}_t^j$. Denote the loss of the classifier with

threshold $\theta$ on the training set $\mathcal{S}$ as $\ell(\theta, \mathcal{S})$, the acceptance rate of group $s$ at $t$ under threshold $\theta$ as $A_\theta^{st}$, and the proportion of agents in group $s$ as $p_s$. The decision-maker selects $\widetilde{\theta}_t^i, \widetilde{\theta}_t^j$ that satisfy the following constrained optimization:

$$\min_{\{\theta^i, \theta^j\}} p_i \ell(\theta^i, \mathcal{S}_t^i) + p_j \ell(\theta^j, \mathcal{S}_t^j)$$

$$\text{s.t. } A_{\theta^i}^{it} = A_{\theta^j}^{jt}$$

To summarize, at $t$, we first train $f_t^i, f_t^j$ with $\mathcal{S}_t^i, \mathcal{S}_t^j$, then tune the thresholds to ensure the new classifiers are fair on the new agents coming at $t$ while optimizing the training loss. WLOG, we assume $i$ is the advantaged group and $j$ is the disadvantaged group, where $A_{\theta_t^i}^{it} > A_{\theta_t^j}^{jt}$. We can have the relationships between the fair optimal thresholds and the original optimal thresholds as follows.

**Proposition D.2.** *If $P_{XY}^{ti}, P_{XY}^{tj}$ are continuous, then $\widetilde{\theta}_t^i \geqslant \theta_t^i$ and $\widetilde{\theta}_t^j \leqslant \theta_t^j$.*

We prove Prop. D.2 in App. G.9. Note that since we need $P_{XY}^{ti}, P_{XY}^{tj}$ to be continuous, Prop. D.2 is only compatible with the noisy best response (Jagadeesan et al., 2021). The noisy best response can follow the setting specified in App. F.5 where the agents only know a noisy version of classification outcomes: $\widehat{f}_t(x) = f_t(x) + \epsilon$ and $\epsilon$ is a Gaussian noise.

With Prop. D.2, we now consider the new dynamics under the fairness interventions (i.e., applying $\widetilde{f}_t^s$ instead of $f_t^s$), where we can define the qualification rates of group $i$ and group $j$, and acceptance rate of both groups at time $t$ as $\widetilde{q}_{i,t}, \widetilde{q}_{j,t}, \widetilde{a}_t$ accordingly.

We first show that under the conditions of Thm. 3.5, compared to the case without fairness intervention, enforcing fairness constraint at each round will cause the advantaged group to improve their qualifications while harming the qualifications of the disadvantaged group.

**Proposition D.3.** *Under the conditions in Thm. 3.5, the fairness intervention at round $t$ can harm the qualification rate for the disadvantaged group $j$ while benefiting the qualification rate for the advantaged group $i$. Specifically, denote $q'_{s,t+1}$ as the qualification rate of group $s$ at $t+1$ if the original optimal threshold $\theta_t^s$ is applied. Then we have $\widetilde{q}_{i,t+1} > q'_{i,t+1}$ but $\widetilde{q}_{j,t+1} < q'_{j,t+1}$.*

Prop. D.3 can be proved by combining Prop. D.2 with the two convexity conditions in Thm. 3.5. Moreover, we can also derive the dynamics of $\widetilde{a}_t, \widetilde{q}_{i,t}, \widetilde{q}_{j,t}$ as follows.

**Proposition D.4.** *Under fairness intervention, the acceptance rates of both groups increase over time, i.e., $\widetilde{a}_t < \widetilde{a}_{t+1}$. Moreover, under the conditions of Thm. 3.5, $\widetilde{q}_{s,t} < \widetilde{q}_{s,t+1}$ for any group $s$.*

These results show that although we can maintain equal acceptance rates among different groups throughout the retraining process, the dynamics of acceptance rates and qualification rates for both groups cannot be stabilized under fairness interventions. However, if applying fairness intervention on the *refined retraining process*, we can obtain stable and fair dynamics.

**Fairness intervention at each round with refined retraining process.** In Prop. D.6, we already prove that the *refined retraining process* will not break the conditional probability $D_{Y|X}^o$. Therefore, the optimal classifiers (without fairness interventions) are always stable regardless of the best response of strategic agents, making the fair classifiers $\widetilde{f}_t^s$ also stable as illustrated in Prop. D.5.

**Proposition D.5.** *If applying fairness intervention at each round with refined retraining process, $\widetilde{a}_t$ will stay stable.*

Prop. D.5 provides a solution to produce fair models with stable dynamics.

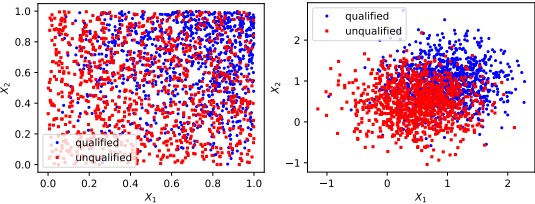

Figure 8: Visualization of distribution: Uniform data (left) and Gaussian data (right)

### D.3 ADDITIONAL DISCUSSIONS ON *refined retraining process*

We provide the following proposition to illustrate how the *refined retraining process* leverages a *probabilistic sampler* to stabilize the dynamics.

**Proposition D.6.** *If the decision-maker uses a probabilistic sampler $\Phi_t(x) = D_{Y|X}^{t-1}(1|x)$ to produce model-annotated samples, $D_{Y|X}^t = D_{Y|X}^o$.*

The proof details are in App. G.8. Prop. D.6 illustrates the underlying conditional distribution $D_{Y|X}^t$ is expected to be the same as $D_{Y|X}^o$, meaning that the classifier $f_t$ always learns the distribution of *human-annotated* data, thereby only preserving the systematic bias. However, there is no way to deal with the systematic bias in *refined retraining process*.

# E    MAIN EXPERIMENTS FOR OTHER DATASETS

We provide results on a Uniform dataset and a real dataset (Quinlan, 2017) with settings similar to Sec. 5.

**Uniform data.** All settings are similar to the Gaussian dataset except that $P_X$ and $P_{Y|X}(1|x)$ change as shown in Table 2.

We first verify the results in Sec. 3 by illustrating the dynamics of $a_t, q_t, \Delta_t$ for both groups (Fig. 9a). Since our analysis neglects the algorithmic bias and the evolution results are in expectation, we perform $n = 100$ independent runs of

Table 2: Gaussian Dataset Setting

| $P_{Xk}(x_k)$ | $P_{Y|X}(1|x)$ | $n, r, T, q_0$ |
|---|---|---|
| $\mathcal{U}(0,1)$ | $0.5 \cdot (x_1 + x_2)$ | $100, 0.05, 15, 0.5$ |

experiments for every parameter configuration and show the averaged outcomes. The results are consistent with Thm. 3.3, 3.5 and 3.6: (i) acceptance rate $a_t$ (red curves) increases monotonically; (ii) qualification rate $q_t$ decreases monotonically starting from $t = 1$ (since strategic agents only best respond from $t = 1$); (iii) classifier bias $\Delta_t$ evolves differently for different groups and it may reach the minimum after a few rounds of retraining. Next, we verify whether the *early stopping* mechanism of the retraining process proposed in Thm. 4.1 can promote fairness. Fig. 9b shows that the decision-maker attains almost *perfect* fairness at $t = 7$. However, as discussed in Sec. 4, although fairness can be enhanced, it only ensures both groups have a similar classifier bias $\Delta_t$ but cannot reduce such bias.

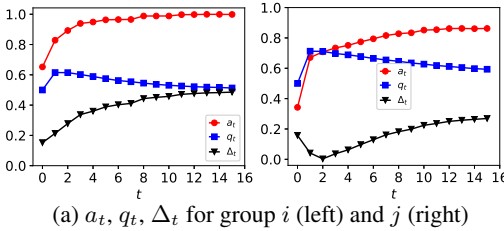
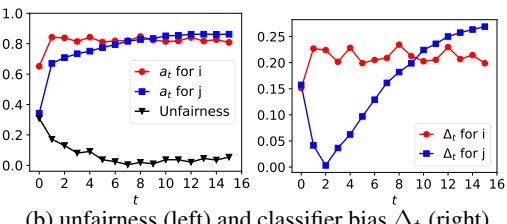

(a) $a_t, q_t, \Delta_t$ for group $i$ (left) and $j$ (right)    (b) unfairness (left) and classifier bias $\Delta_t$ (right)

Figure 9: Dynamics of $a_t, q_t, \Delta_t$ and unfairness $|a_t^i - a_t^j|$ on Gaussian dataset.

Next, we present the results of a set of complementary experiments on real data (Quinlan, 2017) where we directly fit $P_{X|Y}$ with Beta distributions. The fitting results slightly violate Assumption 3.2. Also $D_X^o$ is not equal to $P_X$. More importantly, logistic models cannot fit these distributions well and produce non-negligible algorithmic bias, thereby violating Assumption 3.1. The following experiments demonstrate how the dynamics change when situations are not ideal.

Table 3: Description of credit approval dataset

| Settings | Group $i$ | Group $j$ |
|---|---|---|
| $P_{X_1|Y}(x_1|1)$ | $Beta(1.37, 3.23)$ | $Beta(1.73, 3.84)$ |
| $P_{X_1|Y}(x_1|0)$ | $Beta(1.50, 4.94)$ | $Beta(1.59, 4.67)$ |
| $P_{X_2|Y}(x_2|1)$ | $Beta(0.83, 2.83)$ | $Beta(0.66, 2.50)$ |
| $P_{X_2|Y}(x_2|0)$ | $Beta(0.84, 5.56)$ | $Beta(0.69, 3.86)$ |
| $n, r, T, q_0$ | $50, 0.05, 15, 0.473$ | $50, 0.05, 15, 10$ |

**Credit approval dataset (Quinlan, 2017).** We consider credit card applications and adopt the data in UCI Machine Learning Repository processed by Dua & Graff (2017). The dataset includes features of agents from two social groups $i, j$ and their labels indicate whether the credit application is successful. We first preprocess the dataset by normalizing and only keeping a subset of features (two continuous $X_1, X_2$) and labels, then we fit conditional distributions $P_{X_k|Y}$ for each group using Beta distributions (Fig. 10) and calculate prior-best-response qualification rates $q_0^i, q_0^j$ from the dataset. The details are summarized in Table 3. All other parameter settings are the same as the ones of synthetic datasets in Sec. 5.

We first illustrate the dynamics of $a_t, q_t, \Delta_t$ for both groups under different $r$. The results are shown in Fig. 11 and are approximately aligned with Thm. 3.3, 3.5 and 3.6: (i) acceptance rate $a_t$ (red curves) has increasing trends; (ii) qualification rate $q_t$ (blue curves) decreases starting from $t = 1$ (since strategic agents only best respond from $t = 1$); (iii) classifier bias $\Delta_t$ (black curves) evolve differently for different groups. Next, we show the evolutions of unfairness and $\Delta_t$ in Fig. 12. Though the dynamics are still approximately aligned with the theoretical results, the changes are not smooth. However, this is not surprising because several assumptions are violated, and the overall trends still stay the same.

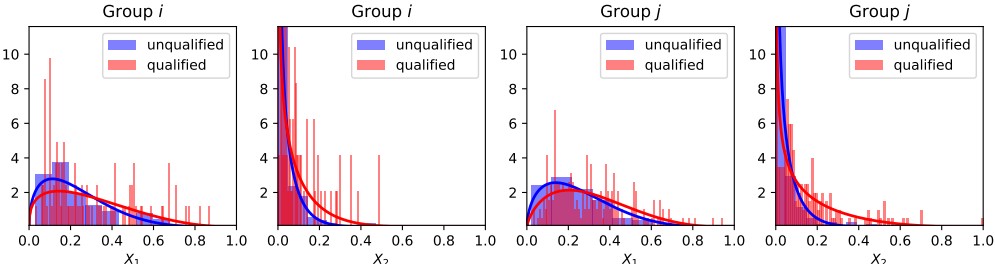

Figure 10: Visualization of distribution for Credit Approval dataset.

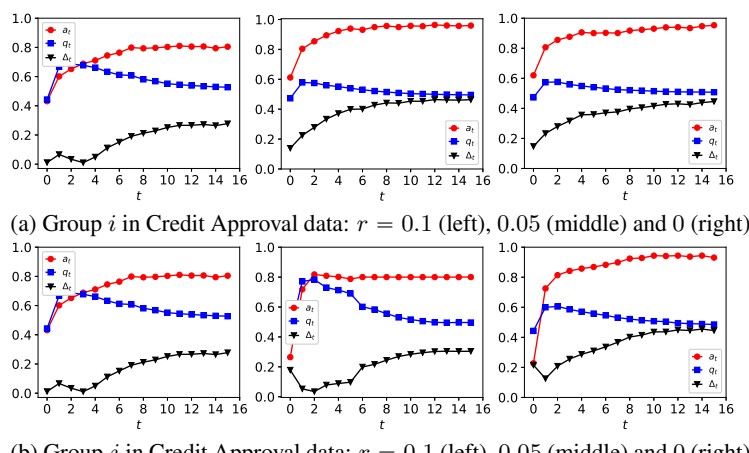

(a) Group $i$ in Credit Approval data: $r = 0.1$ (left), 0.05 (middle) and 0 (right)

(b) Group $j$ in Credit Approval data: $r = 0.1$ (left), 0.05 (middle) and 0 (right)

Figure 11: Dynamics of $a_t, q_t, \Delta_t$ for Credit Approval dataset

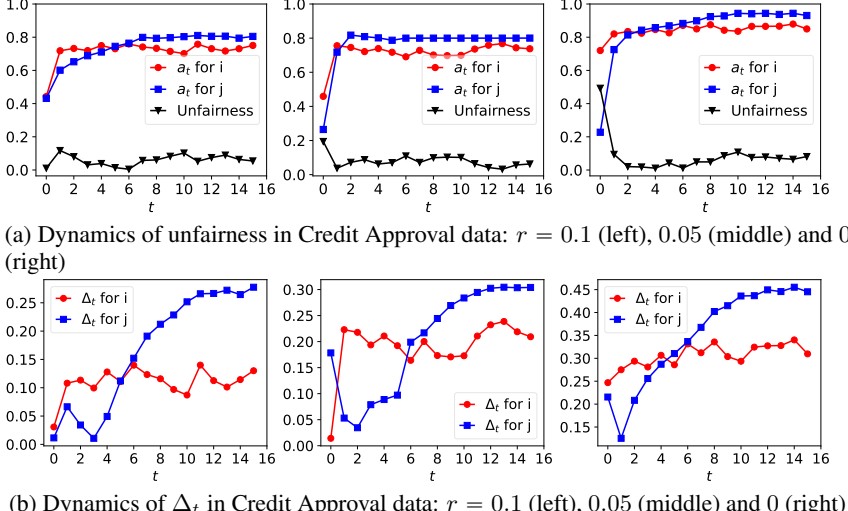

(a) Dynamics of unfairness in Credit Approval data: $r = 0.1$ (left), 0.05 (middle) and 0 (right)

(b) Dynamics of $\Delta_t$ in Credit Approval data: $r = 0.1$ (left), 0.05 (middle) and 0 (right)

Figure 12: Dynamics of unfairness and $\Delta_t$ for Credit Approval dataset

# F  ADDITIONAL RESULTS ON SYNTHETIC/SEMI-SYNTHETIC DATASETS

In this section, we provide comprehensive experimental results conducted on two synthetic datasets and one semi-synthetic dataset mentioned in Sec. 5 of the main paper. Specifically, App. F.1 gives the details of experimental setups; App. F.2 demonstrates additional results to verify Theorem 3.3 to Theorem 3.6 under different $r$ (i.e., ratios of human-annotated examples available at each round).

The section also gives results on how $a_t, q_t, \Delta_t$ change under a long time horizon or when there is no systematic bias; App. F.3 further demonstrates the results under *refined retraining process*; App. F.4 provides fairness dynamics under various values of $r$ on different datasets; App. F.5 illustrates how the results in the main paper still hold when strategic agents have noisy best responses; App. F.6 compares the situations when agents are non-strategic with the ones when they are strategic, demonstrating how agents' strategic behaviors produce more extreme dynamics of $a_t, q_t, \Delta_t$.

## F.1 ADDITIONAL EXPERIMENTAL SETUPS

Generally, we run all experiments on a MacBook Pro with Apple M1 Pro chips, memory of 16GB and Python 3.9.13. All experiments are randomized with seed 42 to run $n$ rounds. Error bars are provided in App. H. All experiments train $f_t$ with a logistic classifier using SGD as its optimizer. Specifically, we use `SGDClassifier` with `logloss` to fit models.

**Synthetic datasets.** The basic description of synthetic datasets 1 and 2 is shown in Sec. 5 and App. E. We further provide the visualizations of their distributions in Fig. 8.

**German Credit dataset.** There are 2 versions of the German Credit dataset according to UCI Machine Learning Database (Hofmann, 1994), and we are using the one where all features are numeric. Firstly, we produce the sensitive features by ignoring the marital status while only focusing on sex. Secondly, we use `MinMaxScaler` to normalize all features. The logistic model itself can satisfy Assumption 3.2 with minimal operations: if feature $i$ has coefficients smaller than 0, then just negate it and the coefficients will be larger than 0 and satisfy the assumption.

## F.2 ADDITIONAL RESULTS TO VERIFY THM. 3.3 TO THM. 3.6

**Dynamics under different $r$.** Although experiments in Sec. 5 and App. E already demonstrate the validity of Thm. 3.3, Prop. 3.4 and Thm. 3.6, the ratio $r$ of human-annotated examples is subject to change in reality. Therefore, we first provide results for $r \in \{0, 0.05, 0.1, 0.3\}$ in all 3 datasets. $r$ only has small values since human-annotated examples are likely to be expensive to acquire. Fig. 15 shows all results under different $r$ values. Specifically, Fig. 15a and 15b show results for synthetic dataset 1, Fig. 15c and 15d show results for synthetic dataset 2, while Fig. 15e and 15f show results for German Credit data. On every row, $r = 0.3, 0.1, 0.05, 0$ from the left to the right. All figures demonstrate the robustness of the theoretical results, where $a_t$ always increases and $q_t$ decreases starting from $t = 1$. $\Delta_t$ also has different dynamics as specified in Thm. 3.6.

**Dynamics under a long time horizon to verify Prop. 3.4.** Prop. 3.4 demonstrates that when $r$ is small enough, $a_t$ will increase towards 1. Therefore, we provide results in all 3 datasets when $r = 0$ to see whether $a_t$ increases to be close to 1. As Fig. 13 shows, $a_t$ is close to 1 after tens of rounds, validating Prop. 3.4.

**Dynamics under a different $B$.** Moreover, individuals may incur different costs to alter different features, so we also provide the dynamics of $a_t, q_t, \Delta_t$ when the cost matrix $B = \begin{bmatrix} 3 & 0 \\ 0 & 6 \end{bmatrix}$ in two synthetic datasets. Fig. 14 shows the differences in costs of changing different features do not affect the theoretical results.

**Dynamics when all samples are human-annotated.** Though this is unlikely to happen under the Strategic Classification setting as justified in the main paper, we provide an illustration when all training examples are *human-annotated* (i.e., $r = 1$) when humans systematically overestimate the qualification in both synthetic datasets. Theoretically, the difference between $a_t$ and $q_t$ should be relatively consistent, which means $\Delta_t$ is only due to the systematic bias. Fig. 13d verifies this.

## F.3 ADDITIONAL RESULTS ON *refined retraining process*

In this section, we provide more experimental results demonstrating how *refined retraining process* stabilizes the dynamics of $a_t, q_t, \Delta_t$ but still preserves the systematic bias. Specifically, we produce plots similar to Fig. 15 in Fig. 16, but the only difference is that we use probabilistic samplers for model-annotated examples. From Fig. 16, it is obvious the deviations of $a_t$ from $q_t$ have the same directions and approximately the same magnitudes as the systematic bias.

### F.4 ADDITIONAL RESULTS ON FAIRNESS

In this section, we provide additional results on the dynamics of unfairness and classifier bias under different $r$ (the same settings in App. F.2). From Fig. 17a, 17c and 17e, we can see unfairness reaches a minimum in the middle of the retraining process, suggesting the earlier stopping of retraining brings benefits. From Fig. 17b, 17d and 17f, we can see $\Delta_t$ for the disadvantaged group $j$ reaches a minimum in the middle of the retraining process, but generally not at the same time when unfairness reaches a minimum.

### F.5 ADDITIONAL RESULTS ON NOISY BEST RESPONSES

Following the discussion in Sec. 5, we provide dynamics of $a_t, q_t, \Delta_t$ of both groups under different $r$ similar to App. F.2 but when the agents have noisy knowledge. The only difference is that the agents' best responses are noisy in that they only know a noisy version of classification outcomes: $\widetilde{f}_t(x) = f_t(x) + \epsilon$, where $\epsilon$ is a Gaussian noise with mean $0$ and standard deviation $0.1$. Fig.18 shows that Thm. 3.3 to Thm. 3.6 are still valid.

### F.6 COMPARISONS BETWEEN STRATEGIC AND NON-STRATEGIC SITUATIONS

In this section, we show the absence of strategic behaviors may result in much more consistent dynamics of $a_t, q_t, \Delta_t$ as illustrated in Fig. 19. Thereby demonstrating the importance of studying the retraining process when strategic behaviors are present.

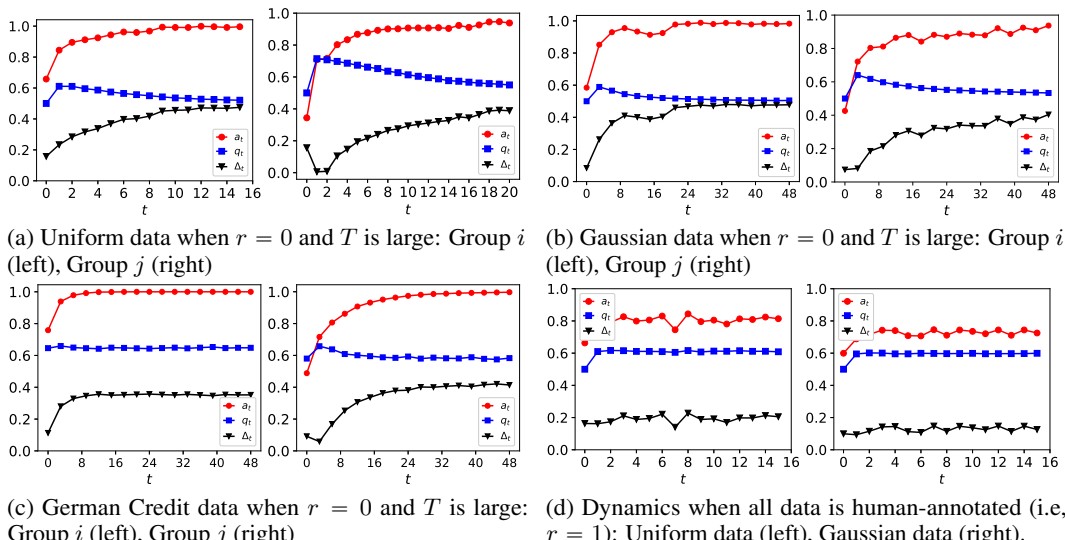

(a) Uniform data when $r = 0$ and $T$ is large: Group $i$ (left), Group $j$ (right)

(b) Gaussian data when $r = 0$ and $T$ is large: Group $i$ (left), Group $j$ (right)

(c) German Credit data when $r = 0$ and $T$ is large: Group $i$ (left), Group $j$ (right)

(d) Dynamics when all data is human-annotated (i.e, $r = 1$): Uniform data (left), Gaussian data (right).

Figure 13: Dynamics of $a_t, q_t, \Delta_t$ on all datasets when $r = 0$ and $T$ is large or when all examples are annotated by humans.

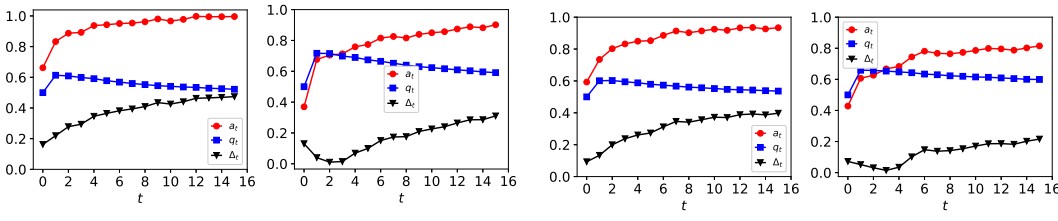

(a) Uniform data with a different $B$ : Group $i$ (left), Group $j$ (right)

(b) Gaussian data with a different $B$ : Group $i$ (left), Group $j$ (right)

Figure 14: Dynamics of $a_t, q_t, \Delta_t$ on synthetic datasets. Except $B$, all other settings are the same as the main experiments in Sec. 5.

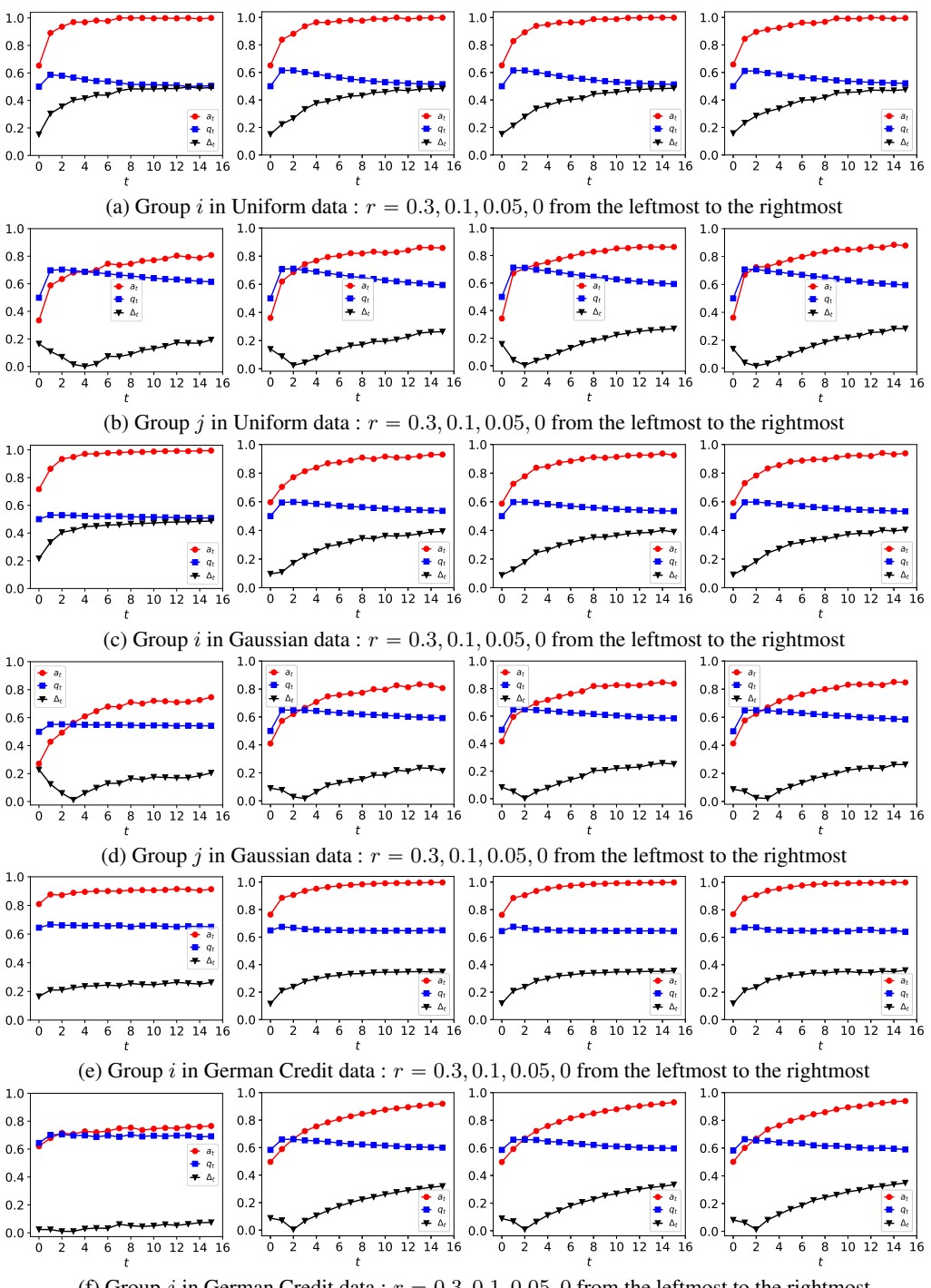

(a) Group $i$ in Uniform data : $r = 0.3, 0.1, 0.05, 0$ from the leftmost to the rightmost

(b) Group $j$ in Uniform data : $r = 0.3, 0.1, 0.05, 0$ from the leftmost to the rightmost

(c) Group $i$ in Gaussian data : $r = 0.3, 0.1, 0.05, 0$ from the leftmost to the rightmost

(d) Group $j$ in Gaussian data : $r = 0.3, 0.1, 0.05, 0$ from the leftmost to the rightmost

(e) Group $i$ in German Credit data : $r = 0.3, 0.1, 0.05, 0$ from the leftmost to the rightmost

(f) Group $j$ in German Credit data : $r = 0.3, 0.1, 0.05, 0$ from the leftmost to the rightmost

Figure 15: Dynamics of $a_t, q_t, \Delta_t$ on all datasets.

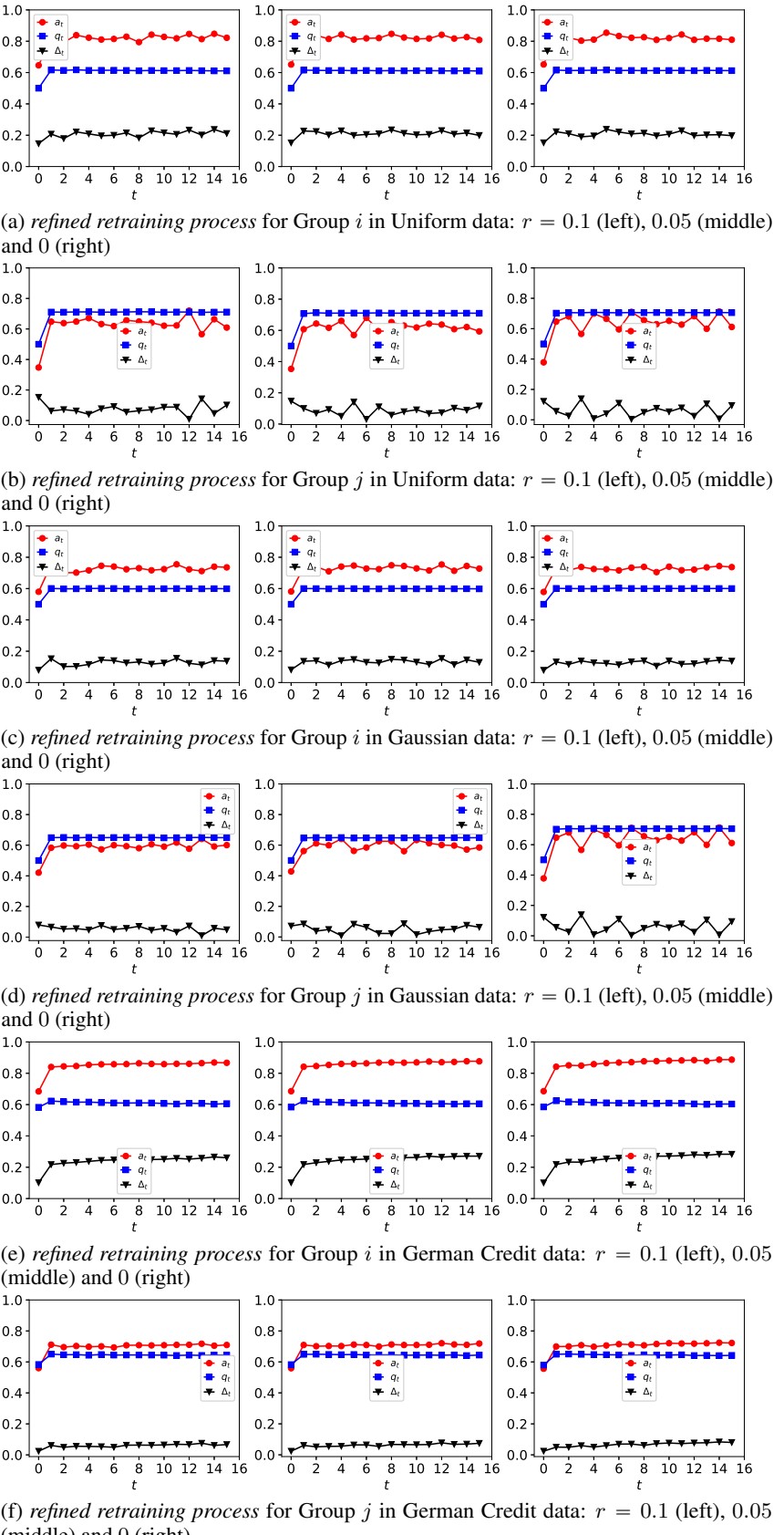

(a) *refined retraining process* for Group $i$ in Uniform data: $r = 0.1$ (left), $0.05$ (middle) and $0$ (right)

(b) *refined retraining process* for Group $j$ in Uniform data: $r = 0.1$ (left), $0.05$ (middle) and $0$ (right)

(c) *refined retraining process* for Group $i$ in Gaussian data: $r = 0.1$ (left), $0.05$ (middle) and $0$ (right)

(d) *refined retraining process* for Group $j$ in Gaussian data: $r = 0.1$ (left), $0.05$ (middle) and $0$ (right)

(e) *refined retraining process* for Group $i$ in German Credit data: $r = 0.1$ (left), $0.05$ (middle) and $0$ (right)

(f) *refined retraining process* for Group $j$ in German Credit data: $r = 0.1$ (left), $0.05$ (middle) and $0$ (right).

Figure 16: Illustrations of *refined retraining process* on all 3 datasets.

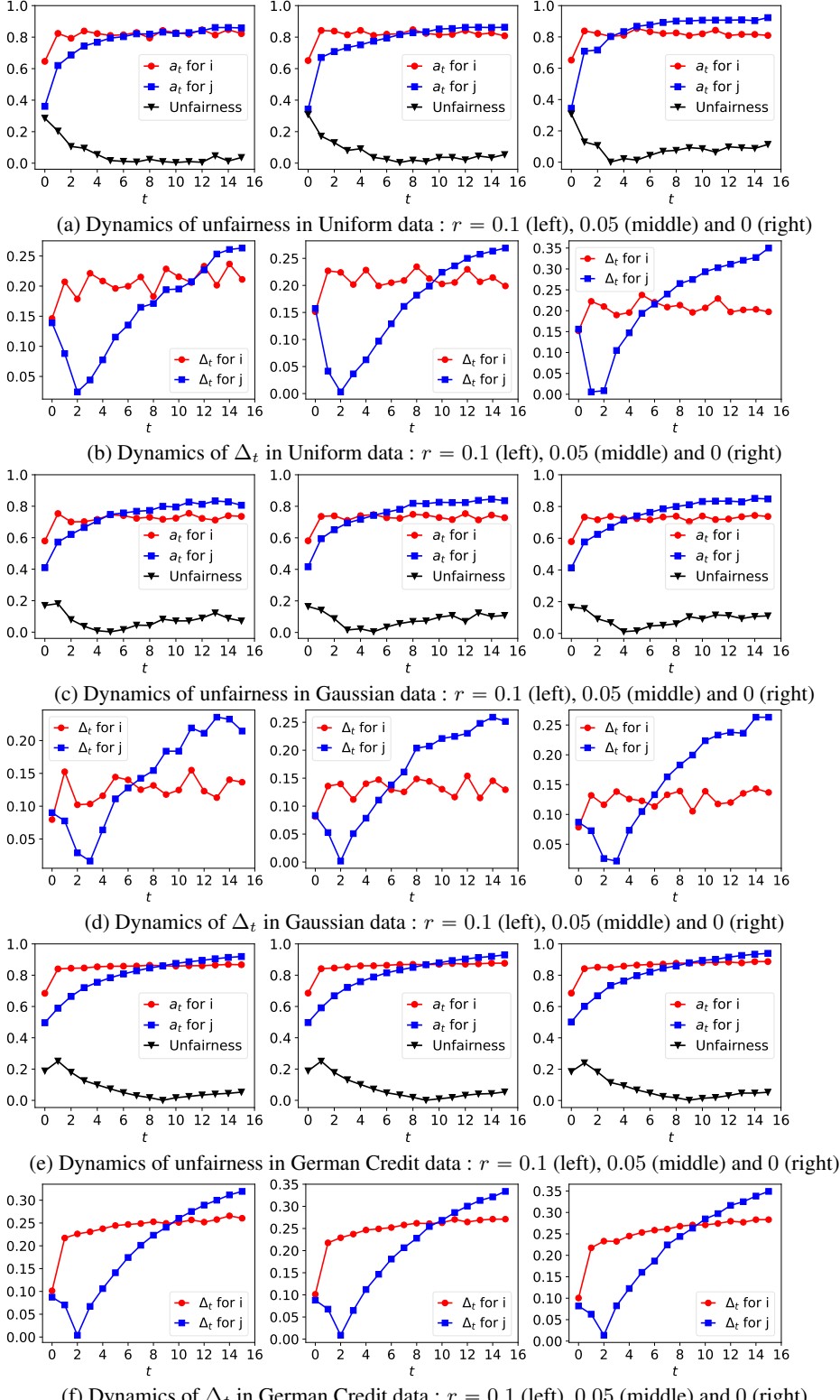

(a) Dynamics of unfairness in Uniform data : $r = 0.1$ (left), $0.05$ (middle) and $0$ (right)

(b) Dynamics of $\Delta_t$ in Uniform data : $r = 0.1$ (left), $0.05$ (middle) and $0$ (right)

(c) Dynamics of unfairness in Gaussian data : $r = 0.1$ (left), $0.05$ (middle) and $0$ (right)

(d) Dynamics of $\Delta_t$ in Gaussian data : $r = 0.1$ (left), $0.05$ (middle) and $0$ (right)

(e) Dynamics of unfairness in German Credit data : $r = 0.1$ (left), $0.05$ (middle) and $0$ (right)

(f) Dynamics of $\Delta_t$ in German Credit data : $r = 0.1$ (left), $0.05$ (middle) and $0$ (right)

Figure 17: Dynamics of unfairness and $\Delta_t$ of all datasets.

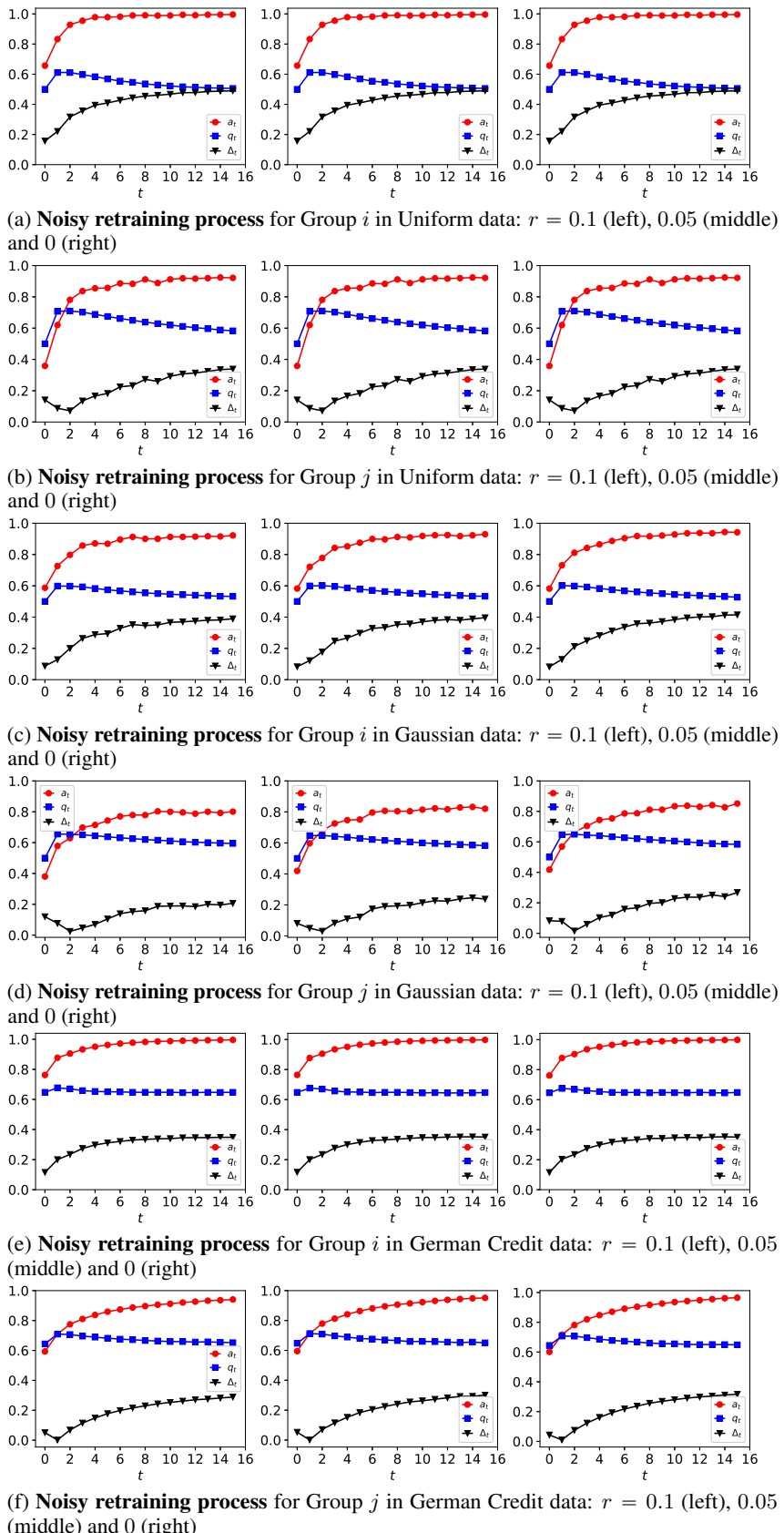

(a) **Noisy retraining process** for Group $i$ in Uniform data: $r = 0.1$ (left), 0.05 (middle) and 0 (right)

(b) **Noisy retraining process** for Group $j$ in Uniform data: $r = 0.1$ (left), 0.05 (middle) and 0 (right)

(c) **Noisy retraining process** for Group $i$ in Gaussian data: $r = 0.1$ (left), 0.05 (middle) and 0 (right)

(d) **Noisy retraining process** for Group $j$ in Gaussian data: $r = 0.1$ (left), 0.05 (middle) and 0 (right)

(e) **Noisy retraining process** for Group $i$ in German Credit data: $r = 0.1$ (left), 0.05 (middle) and 0 (right)

(f) **Noisy retraining process** for Group $j$ in German Credit data: $r = 0.1$ (left), 0.05 (middle) and 0 (right)

Figure 18: Illustrations of **noisy retraining process** on all 3 datasets.

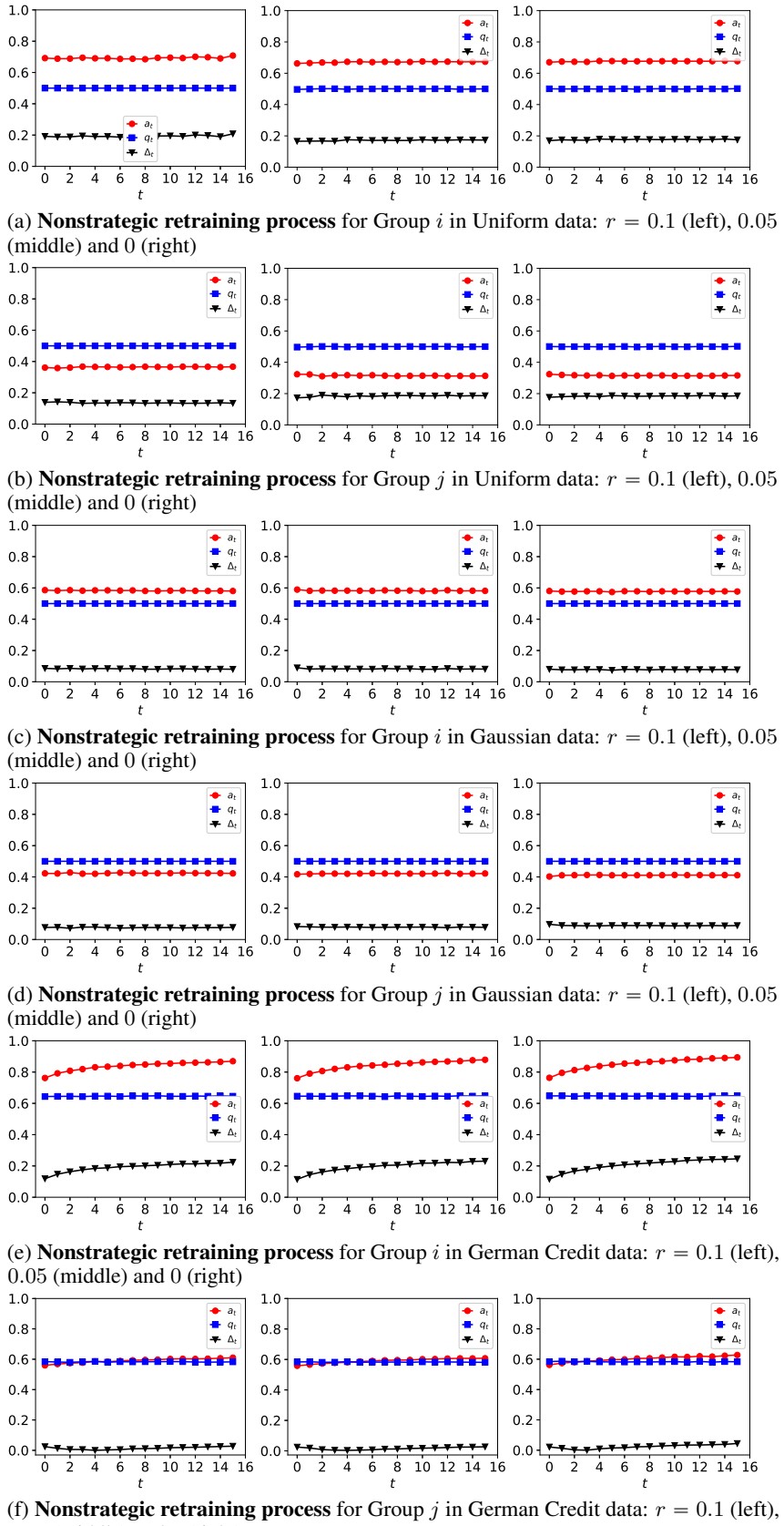

(a) **Nonstrategic retraining process** for Group $i$ in Uniform data: $r = 0.1$ (left), 0.05 (middle) and 0 (right)

(b) **Nonstrategic retraining process** for Group $j$ in Uniform data: $r = 0.1$ (left), 0.05 (middle) and 0 (right)

(c) **Nonstrategic retraining process** for Group $i$ in Gaussian data: $r = 0.1$ (left), 0.05 (middle) and 0 (right)

(d) **Nonstrategic retraining process** for Group $j$ in Gaussian data: $r = 0.1$ (left), 0.05 (middle) and 0 (right)

(e) **Nonstrategic retraining process** for Group $i$ in German Credit data: $r = 0.1$ (left), 0.05 (middle) and 0 (right)

(f) **Nonstrategic retraining process** for Group $j$ in German Credit data: $r = 0.1$ (left), 0.05 (middle) and 0 (right)

Figure 19: Illustrations of **nonstrategic retraining process** on all 3 datasets.

# G DERIVATIONS AND PROOFS

## G.1 DERIVATION OF EQUATION 3

To derive $\bar{q}_t := \mathbb{E}_{\mathcal{S}_t}[Q(\mathcal{S}_t)]$, we first refer to equation 2 to get $|\mathcal{S}_t| = (t+1)N + tK$. Then, by the law of total probability, the expected qualification rate of $\mathcal{S}_t$ equals to the weighted sum of the expected qualification rate of $\mathcal{S}_{t-1}, S_{o,t-1}$ and the expectation of $f_{t-1}(x)$ over $\mathcal{S}_{m,t-1}$ as follows:

$$\frac{tN+(t-1)K}{(t+1)N+tK}\mathbb{E}_{\mathcal{S}_{t-1}}[Q(\mathcal{S}_{t-1})] + \frac{N}{(t+1)N+tK} + \mathbb{E}_{\mathcal{S}_{t-1}}[A(f_{t-1}, P^{t-1}] + \frac{K}{(t+1)N+tK}\mathbb{E}_{\mathcal{S}_{o,t-1}}[Q(\mathcal{S}_{o,t-1})]$$

The second expectation is exactly the definition of $a_{t-1}$. Moreover, note that $Q(\mathcal{S}_{o,t-1}) = Q(\mathcal{S}_{o,0}) = Q(\mathcal{S}_0) = \bar{q}_0$ for any $t > 0$, the third expectation is exactly $\bar{q}_0$, so the above equation is exactly equation 3.

## G.2 DERIVATION OF FACTORS INFLUENCING $a_t, q_t$

As stated in Sec.2, we can get the factors influencing the evolution of $a_t, q_t$ by finding all sources affecting $P^t_{XY}$ and the expectation of $f_t(x)$ over $P^t_X$.

We first work out the sources influencing $f_t$ and the expectation of $f_t(x)$ over $P^t(X)$:

- $\bar{q}_t$: since $f_t$ is trained with $\mathcal{S}_t$, $\bar{q}_t$ is a key factor influencing the classifier.
- $\delta(D^t, \mathcal{F})$ : note that as retaining goes on, $|\mathcal{S}^t|$ is large enough. Thus, if $f_t$ fits $\mathcal{S}_t$ well with a low error, the expectation of $f_t(x)$ over $D^t_X$ should be close to $\bar{q}_t$, and all the difference is due to the algorithmic bias. This claim is supported by Chapter 5.2 of Shalev-Shwartz & Ben-David (2014), where the error of an $ERM$ predictor is decomposed into the approximation error and the estimation error. When the sample size is large and $|F|$ is finite, the estimation error will be arbitrarily small, leaving only the approximation error. The approximation error measures how well $F$ can approximate $D^t$, which is exactly the algorithmic bias $\delta(D^t, \mathcal{F})$.
- $\delta^t_{BR}$: we now know factors influencing the expectation of $f_t(x)$ over $D^t_X$, then the only left factor is the ones accounting for the difference between $D^t_X$ and $P^t_X$. Note that only the best responses of agents can change the marginal distribution $P_X$. We then denote it as $\delta^t_{BR}$.

Then, with $P_{XY}$ known, $P^t_{XY}$ is only influenced by $f_{t-1}$ (i.e., the agents' best responses to the classifier at $t-1$). $f_{t-1}$ is also dependent on the above factors. So we get all factors.

## G.3 PROOF OF THEOREM 3.3

Let us begin by proving the following lemma:

**Lemma G.1.** *Assume $t \geq 2$ and the following conditions hold: (i) $\bar{q}_t > \bar{q}_{t-1} \geq \bar{q}_{t-2}$; (ii) $\forall x \in X$, $D^t_{Y|X}(1|x) \geq D^{t-1}_{Y|X}(1|x)$; (iii) $\forall x, \bar{f}_{t-1}(x) \geq \bar{f}_{t-2}(x)$. Let $\bar{f}_{t-1} = \mathbb{E}_{\mathcal{S}_{t-1} \sim D^{t-1}_{XY}}[f_{t-1}], \bar{f}_t = \mathbb{E}_{\mathcal{S}_t \sim D^t_{XY}}[f_t]$, we have the following results:*

*1. $\forall x, \bar{f}_t(x) \geq \bar{f}_{t-1}(x)$.*

*2. There exists a non-zero measure subset of $x$ values that satisfies the strict inequality.*

**Proof.** We first prove (i). Note that we ignore the algorithm bias in Assumption 3.1, so $f_t, f_{t-1}$ are expected to model the conditional distributions $D^t_{Y|X}, D^{t-1}_{Y|X}$ well. Therefore, $\bar{f}_{t-1}$ (resp. $\bar{f}_t$) outputs 1 if $D^{t-1}_{Y|X}(1|x)$ (resp. $D^t_{Y|X}(1|x)$) is larger than some threshold $\theta$. Then, according to the above condition (ii), $\forall \theta$, if $D^{t-1}_{Y|X}(1|x) > \theta$, $D^t_{Y|X}(1|x) \geq D^{t-1}_{Y|X}(1|x) > \theta$. This demonstrates that $\bar{f}_{t-1}(x) = 1$ implies $\bar{f}_t(x) = 1$ and (i) is proved.

Next, according to equation 2, if $\bar{q}_t > \bar{q}_{t-1}$, this means $\mathbb{E}_{y \sim D^t_Y}[y] > \mathbb{E}_{y \sim D^t_Y}[y]$. Since $D^t_Y = D^t_X \cdot D^t_{Y|X}$, **either** there exists at least one non-zero measure subset of $x$ values satisfying $D^t_{Y|X}(1|x) > D^{t-1}_{Y|X}(1|x)$ **or** $D^t_X$ is more "skewed" to the larger values of $x$ (because of monotonic likelihood assumption 3.2). For the second possibility, note that the only possible cause for the

feature distribution in the training dataset $D_X^t$ to gain such a skewness is agents' strategic behaviors. However, since $\bar{f}_{t-1}(x) \geqslant \bar{f}_{t-2}(x)$ always holds, $\bar{f}_{t-1}$ sets a lower admission standard where some $x$ values that are able to best respond to $\bar{f}_{t-2}$ and improve will not best respond to $\bar{f}_{t-1}$, thereby impossible to result in a feature distribution shift to larger $x$ values while keeping the conditional distribution unchanged. Thus, only the first possibility holds, and the lemma is proved. □

Then we prove Theorem 3.3 using mathematical induction to prove a stronger version:

**Lemma G.2.** *When* $t > 1$, $\bar{q}_t > \bar{q}_{t-1}$, $D_{Y|X}^t(1|x) \geqslant D_{Y|X}^{t-1}(1|x)$, $\bar{f}_t(x) \geqslant \bar{f}_{t-1}(x)$, *and finally* $a_t > a_{t-1}$.

**Proof.** $t$ starts from 2, but we need to prove the following claim: $\bar{q}_1$ is "almost equal" to $\bar{q}_0$, so are $D_{Y|X}^1$ and $D_{Y|X}^0$.

Firstly, according to the law of total probability, we can derive $\bar{q}_1$ as follows:

$$\bar{q}_1 = \frac{N}{2N+K} \cdot \bar{q}_0 + \frac{N}{2N+K} \cdot a_0 + \frac{K}{2N+K} \cdot \bar{q}_0 \tag{5}$$

The first and the third element are already multiples of $\bar{q}_0$. Also, we know $D_X^o = P_X$. Then, since $\delta(D^o, \mathcal{F}$ is negligible and agents at $t = 0$ have no chance to best respond, we have $a_0 = \mathbb{E}_{X \sim P_X}[D_{Y|X}^o(1|x)] = \bar{q}_0$. Thus, $a_0$ is also equal to $\bar{q}_0$, and the claim is proved. Still, as $\delta(D^o, \mathcal{F}$ is negligible, $\bar{f}_1$ is the same as $\bar{f}_0$.

**Next, we can prove the lemma by induction**:

1. Base case: Similar to Eq.5, we are able to derive $\bar{q}_2$ as follows:

$$\bar{q}_2 = \frac{2N+K}{3N+2K} \cdot \bar{q}_1 + \frac{N}{3N+2K} \cdot a_1 + \frac{K}{3N+2K} \cdot \bar{q}_0 \tag{6}$$

Based on the claim above, we can just regard $\bar{q}_0$ as $\bar{q}_1$. Then we may only focus on the second term. Since $\bar{q}_1$ and $\bar{q}_0$ are "almost equal" and both distributions should satisfy the monotonic likelihood assumption 3.2, we can conclude $\bar{f}_0, \bar{f}_1$ are "almost identical". Then the best responses of agents to $\bar{f}_0$ will also make them be classified as 1 by $\bar{f}_1$, and this will directly ensure the second term to be $A(f_1, P^1) > A(f^1, P) = A(f^0, P)$. The "larger than" relationship is because strategic best responses at the first round will only enable more agents to be admitted. Thus, the first and the third term stay the same as $\bar{q}_1$ while the second is larger, so we can claim $\bar{q}_2 > \bar{q}_1$. Moreover, the difference between $D_{XY}^2$ and $D_{XY}^1$ are purely produced by the best responses at $t = 1$, which will never decrease the conditional probability of $y = 1$. Thus, $D_{Y|X}^2(1|x) \geqslant D_{Y|X}^1(1|x)$. Together with $\bar{f}_1 = \bar{f}_0$, all three conditions in Lemma G.1 are satisfied. we thereby claim that for every $x$ admitted by $\bar{f}_1$, $\bar{f}_2(x) \geqslant \bar{f}_1(x)$ and there exists some $x$ satisfying the strict inequality. Note that $P^1$ is expected to be the same as $P^2$ since $f_1 = f_0$. Thus, $A(f_2, P^2) > A(f_1, P^2) = A(f_1, P^1)$, which is $a_2 > a_1$. The base case is proved.

2. Induction step: To simplify the notion, we can write:

$$
\begin{aligned}
\bar{q}_t &= \frac{tN + (t-1)K}{(t+1)N + tK} \cdot \bar{q}_{t-1} + \frac{N}{(t+1)N + tK} \cdot a_{t-1} + \frac{K}{(t+1)N + tK} \cdot \bar{q}_0 \\
&= A_t \cdot \bar{q}_{t-1} + B_t \cdot a_{t-1} + C_t \cdot \bar{q}_0
\end{aligned}
$$

Note that $\bar{q}_{t-1}$ can also be decomposed into three terms:

$$
\begin{aligned}
\bar{q}_{t-1} &= \frac{(t-1)N + (t-2)K}{tN + (t-1)K} \cdot \bar{q}_{t-2} + \frac{N}{tN + (t-1)K} \cdot a_{t-2} + \frac{K}{tN + (t-1)K} \cdot \bar{q}_0 \\
&= A_{t-1} \cdot \bar{q}_{t-1} + B_{t-1} \cdot a_{t-2} + C_{t-1} \cdot \bar{q}_0
\end{aligned}
$$

Since the expectation in the second term is just $a_{t-1}$, and we already know $a_{t-1} > a_{t-2}$, we know $B_t \cdot a_{t-1} + C_t \cdot \bar{q}_0 > B_t \cdot a_{t-2} + C_t \cdot \bar{q}_0$. Note that $\frac{B_t}{B_{t-1}} = \frac{C_t}{C_{t-1}}$, we let the ratio be $m < 1$. Then since $B_{t-1} \cdot a_{t-2} + C_{t-1} \cdot \bar{q}_0 > (B_{t-1} + C_{t-1}) \cdot \bar{q}_{t-1}$ due to $\bar{q}_{t-1} > \bar{q}_{t-2}$, we can derive $B_t \cdot a_{t-1} + C_t \cdot \bar{q}_0 > m \cdot (B_{t-1} + C_{t-1}) \cdot \bar{q}_{t-1} = (B_t + C_t) \cdot \bar{q}_{t-1}$. Then $\bar{q}_t > (A_t + B_t + C_t) \cdot \bar{q}_{t-1} = \bar{q}_{t-1}$. The first claim is proved. As $a_{t-1} > a_{t-2}$ and $D^o$ stays the same, any agent will not have a less probability of being qualified in $\mathcal{S}_t$ compared to in $\mathcal{S}_{t-1}$, demonstrating $D_{Y|X}^t(1|x) \geqslant D_{Y|X}^{t-1}(1|x)$ still holds. And similarly, we can apply Lemma G.1 to get $\bar{f}_t(x) \geqslant \bar{f}_{t-1}(x)$ and $a_t > a_{t-1}$. $\qquad\square$

Now we already prove Lemma G.2, which already includes Theorem 3.3. We also want to note here, the proof of Theorem 3.3 does not rely on the initial $\bar{q}_0$ which means it holds regardless of the systematic bias.

### G.4 Proof of Proposition 3.4

We prove the proposition by considering two extreme cases:

(i) When $\frac{K}{N} \to 0$, this means we have no decision-maker annotated sample coming in each round and all new samples come from the deployed model. We prove $\lim_{t\to\infty} a_t = 1$ by contradiction: firstly, by *Monotone Convergence Theorem*, the limit must exist because $a_t > a_{t-1}$ and $a_t < 1$. Let us assume the limit is $\bar{a} < 1$. Then, since $K = 0$, when $t \to \infty$, $\bar{q}_t$ will also approach $\bar{a}$, this means the strategic shift $\delta_{BR}^t$ approaches 0. However, this shift only approaches 0 when all agents are accepted by $f_{t-1}$ because otherwise there will be a proportion of agents benefiting from best responding to $f_{t-1}$ and result in a larger $\bar{q}_{t+1}$ Thus, the classifier at $t+1$ will admit more people and the stability is broken. This means the only possibility is $lim_{t\to\infty} a_{t-1} = 1$ and produces a conflict.

(ii) When $\frac{K}{N} \to \infty$, the problem shrinks to retrain the classifier to fit $D_{XY}^o$, this will make $a_t = a_0$.

Thus, there exists some threshold $\lambda$, when $\frac{K}{N} < \lambda$, $lim_{t\to\infty} a_t = 1$. In practice, the $\lambda$ could be very small. $\qquad\square$

### G.5 Proof of Theorem 3.5

Firstly, since $P_{XY}^t$ differs from $P_{XY}$ only because agents' best respond to $f_t$, we can write $q_t = q_0 + r_t$ where $r_t$ is the difference of qualification rate caused by agents' best responses to $f_t$. Qualitatively, $r_t$ is completely determined by two sources: (i) the proportion of agents who move their features when they best respond; (ii) the increase in the probability of being qualified for each agent. Specifically, each agent that moves its features increases its probability of being qualified from its initial point to a point at the decision boundary of $f_t$. For an agent with initial feature $x$ and improved feature $x^*$, its improvement can be expressed as $U(x) = P_{Y|X}(1|x^*) - P_{Y|X}(1|x)$.

Next, denote the Euclidean distance between $x$ and the decision boundary of $f_t$ as $d_{x,t}$. Noticing that the agents who will choose to move their features to best respond should have distances within a threshold $C$ to the decision boundary no matter where the boundary is (Lemma 2 in Levanon & Rosenfeld (2022)), we can express the total improvement at $t$ as follows:

$$I(t) = \int_{d_{x,t} \leqslant C} P_X(x) \cdot U(x)\, dx \qquad (7)$$

According to the proof of Theorem 3.3, all agents with feature vector $x$ who are admitted by $f_0$ will be admitted by $f_t$ $(t > 0)$, making all agents who possibly improve must belong to $\mathcal{J}$. Since $F_X$ and $P_{Y|X}(1|x)$ are both convex and non-decreasing in $\mathcal{J}$, $P_X(x), U(x)$ are non-decreasing in $\mathcal{J}$. Then note that $f_t$ always lies below $f_{t-1}$, therefore, for each agent who improves at $t$ having feature vector $x_{i,t}$, we can find an agent at $t-1$ with corresponding $x_{i,t-1}$ such that both $P_X(x_{i,t-1}) > P_X(x_{i,t})$ and $U(x_{i,t-1}) > U(x_{i,t})$. This will ensure $I(t) < I(t-1)$. Thus, $q_t$ decreases starting from 1. $\quad\square$

Generally, equation 7 reveals that if $P_X(x) \cdot U(x)$ is convex in $\mathcal{J}$, $q_t$ decreases. This illustrates Prop. 3.5 gives a sufficient condition which is much easier to verify.

### G.6 PROOF OF THEOREM 3.6

- $\mu(D^o, P) \geqslant 0$: according to Def. 2.1, $a_0 = \mathbb{E}_{P_X}[D^o_{Y|X}(1|x)] > \mathbb{E}_{P_X}[P_{Y|X}(1|x)] = Q(P) = q_0$. Now that $a_0 - q_0 \geqslant 0$. Based on Thm. 3.3 and Thm. 3.5, $a_t$ is increasing, while $q_0$ is decreasing, so $\Delta_t$ is always increasing.

- $\mu(D^o, P) < 0$: similarly we can derive $a_0 - q_0 < 0$, so $\Delta_0 = |a_0 - q_0| = q_0 - a_0$. So while $a_0 - q_0$ is still increasing, $\Delta_t$ will first decrease. Moreover, according to Prop. 3.4, if $\frac{K}{N}$ is small enough, $\Delta_t$ will eventually exceed 0 and become larger again. Thus, $\Delta_t$ either decreases or first decreases and then increases. $\qquad\square$

### G.7 PROOF OF THEOREM 4.1

Assume the acceptance rate of group $i$ and group $j$ at time $t$ are $a^i_t, a^j_t$, and the unfairness measured by DP is just $|a^i_t - a^j_t|$ at time $t$. Since the conditions in Prop. 3.4 and Thm. 3.5 are satisfied for both groups, $a^i_t, a^j_t$ will both approach 1 if applying the original retraining process.

At first, $a^i_t < a^j_t$. If the decision-maker applies *refined retraining process* on group $i$, $a^i_t$ will be relatively consistent since the first round of training. But $a^j_t$ keeps increasing. So after some rounds of training, $a^i_t$ will be approximately equal to $a^j_t$. However, if the decision maker keeps retraining, $a^i_t$ will be larger than $a^j_t$ and approach 1, resulting in the unfairness first decreasing and then increasing. Though this implicitly assumes the changes are continuous, the experiments in Sec. 5 verify that the discrete situations indeed display the same nature. $\qquad\square$

### G.8 PROOF OF PROPOSITION D.6

Firstly, $\mathcal{S}_0 = \mathcal{S}_{o,0}$ and $S_1 = \mathcal{S}_0 \cup \mathcal{S}_{o,1} \cup \mathcal{S}_{m,0}$. Obviously, the first two sets are drawn from $D^o_{XY}$. Consider $\mathcal{S}_{m,0}$, since it is now produced by labeling features from $P_X = D^o_X$ with $D^o_{Y|X}$, $\mathcal{S}_{m,0}$ is also drawn from $D^o_{XY}$. Thus, $D^1_{Y|X} = D^o_{Y|X}$.

Then we prove the cases when $t > 1$ using mathematical induction as follows:

1. Base case: We know $\mathcal{S}_2 = \mathcal{S}_1 \cup \mathcal{S}_{o,2} \cup \mathcal{S}_{m,1}$. The first two sets on rhs are drawn from $D^o_{XY}$. The labeing in the third set is produced by $\Phi_1(x) = D^1_{Y|X} = D^o_{Y|X}$. Thus, the base case is proved.

2. Induction step: $\mathcal{S}_t = \mathcal{S}_{t-1} \cup \mathcal{S}_{o,t} \cup \mathcal{S}_{m,t-1}$. Similarly, we only need to consider the third set. But Note that $\Phi_t(x) = D^{t-1}_{Y|X} = D^o_{Y|X}$, the induction step is easily completed. $\qquad\square$

### G.9 PROOF OF PROPOSITION D.2

At round $t$, we know $A^{it}_{\theta^i_t} > A^{jt}_{\theta^j_t}$. To reach demographic parity, the acceptance rates need to be the same, so at least one of the following situations must happen: (i) $\widetilde{\theta}^i_t > \theta^i_t$ and $\widetilde{\theta}^j_t > \theta^j_t$; (ii) $\widetilde{\theta}^i_t < \theta^i_t$ and $\widetilde{\theta}^j_t < \theta^j_t$; (iii) $\widetilde{\theta}^i_t < \theta^i_t$ and $\widetilde{\theta}^j_t < \theta^j_t$. Next we prove that (i) and (ii) cannot be true by contradiction.

Suppose (i) holds, then we can find $\overline{\theta}^j_t < \theta^j_t$ such that $\ell(\overline{\theta}^j_t, \mathcal{S}^j_t) \in \left(\ell(\theta^j_t, \mathcal{S}^j_t), \ell(\widetilde{\theta}^j_t, \mathcal{S}^j_t)\right)$ and $A^{jt}_{\overline{\theta}^j_t} \in \left(A^{jt}_{\theta^j_t}, A^{it}_{\theta^i_t}\right)$. We can indeed find this $\overline{\theta}^j_t$ because $\ell, A$ are continuous w.r.t. $\theta$. Now noticing that $A^{it}_{\widetilde{\theta}^i_t} = A^{jt}_{\overline{\theta}^j_t} > A^{jt}_{\theta^j_t}$ but $A^{jt}_{\overline{\theta}^j_t} < A^{jt}_{\theta^j_t}$, we will know we can find $\overline{\theta}^i_t \in \left(\theta^i_t, \widetilde{\theta}^i_t\right)$ to satisfy demographic parity together with $\overline{\theta}^j_t$. Since $P_{Y|X}(1|x)$ satisfies monotonic likelihood and $\theta^i_t$ is the optimal point, $\ell(\overline{\theta}^i_t, \mathcal{S}^i_t) < \ell(\widetilde{\theta}^i_t, \mathcal{S}^i_t)$ must hold. Thus, $(\overline{\theta}^i_t, \overline{\theta}^j_t)$ satisfy demographic parity and have a lower loss than $(\widetilde{\theta}^i_t, \widetilde{\theta}^j_t)$. Moreover, the pair satisfies (iii), which produces a conflict.

Similarly, we can prove (ii) cannot hold by contradiction, thereby proving (iii) must hold.

## H  ALL PLOTS WITH ERROR BARS

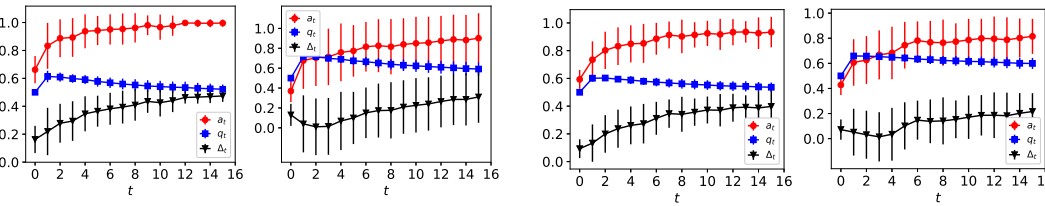

(a) Uniform data with a different $B$ : Group $i$ (left), Group $j$ (right)

(b) Gaussian data with a different $B$ : Group $i$ (left), Group $j$ (right)

Figure 20: Error bar version of Fig. 14.

Although all our theoretical results are expressed in terms of expectation, we provide error bars for all plots in the main paper, App. E and F if randomness is applicable. All following figures demonstrate expectations as well as error bars ($\pm$ 1 standard deviation). Overall, the experiments have reasonable standard errors. However, experiments in the Credit Approval dataset Quinlan (2017) (Fig. 33) incur larger standard errors, which is not surprising because the dataset violates several assumptions. Finally, note that we conduct 50-100 randomized trials for every experiment and we should expect much lower standard errors if the numbers of trials become large.

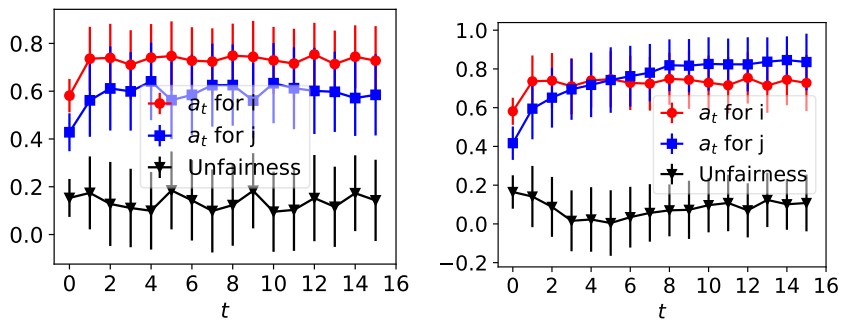

Figure 21: Error bar version of Fig. 4

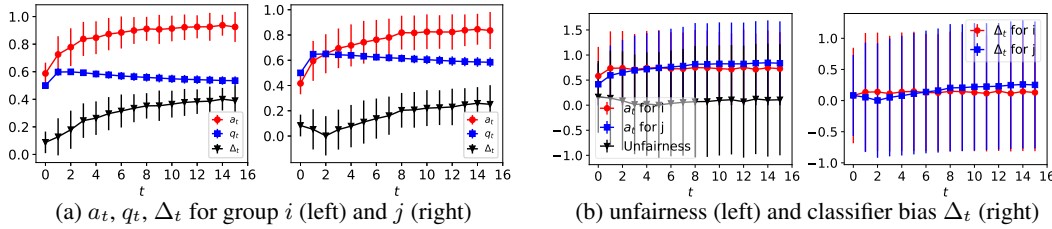

(a) $a_t, q_t, \Delta_t$ for group $i$ (left) and $j$ (right)   (b) unfairness (left) and classifier bias $\Delta_t$ (right)

Figure 22: Error bar version of Fig. 5

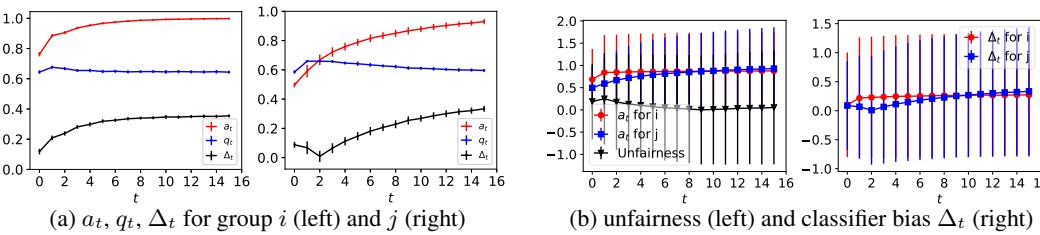

(a) $a_t, q_t, \Delta_t$ for group $i$ (left) and $j$ (right)   (b) unfairness (left) and classifier bias $\Delta_t$ (right)

Figure 23: Error bar version of Fig. 6

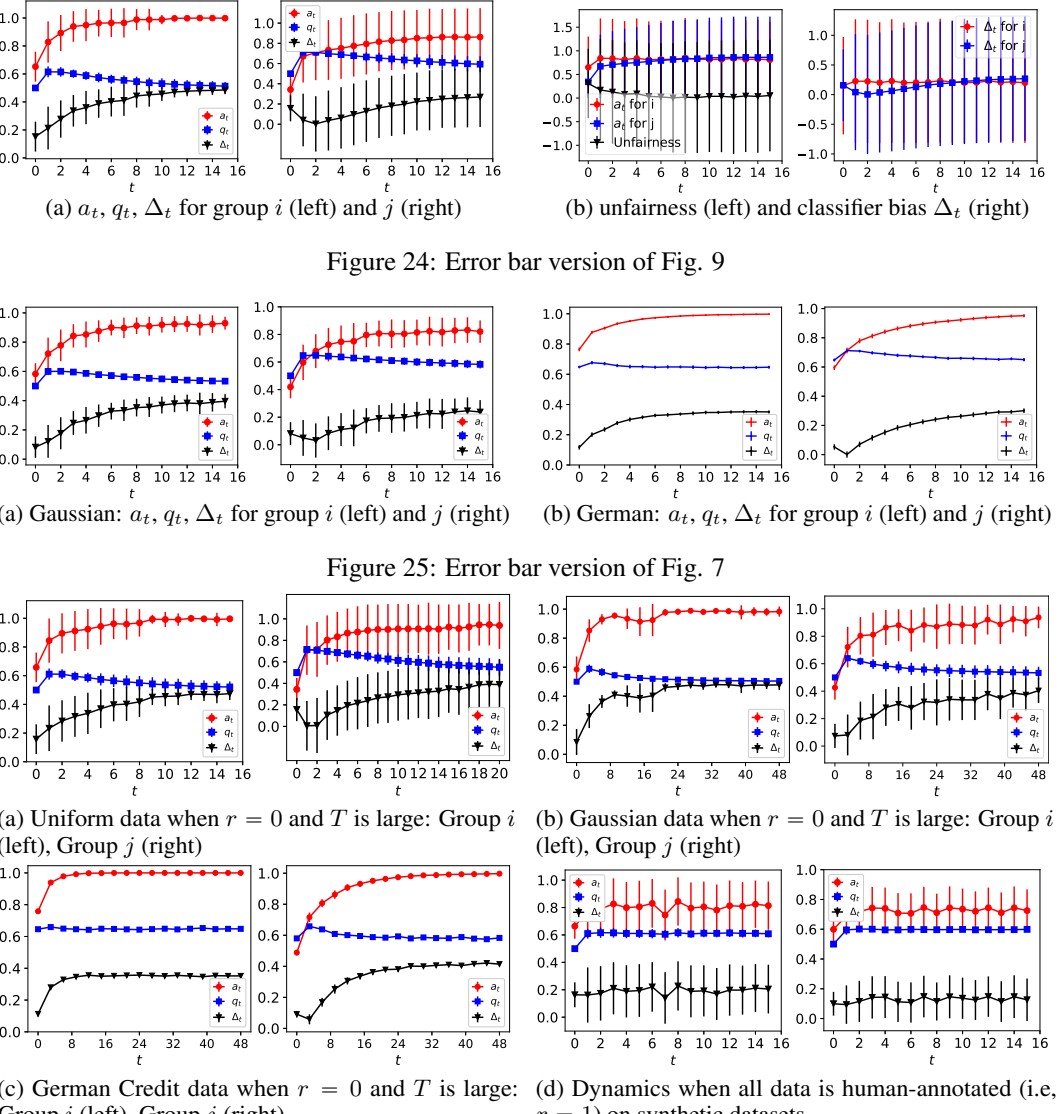

(a) $a_t, q_t, \Delta_t$ for group $i$ (left) and $j$ (right)

(b) unfairness (left) and classifier bias $\Delta_t$ (right)

Figure 24: Error bar version of Fig. 9

(a) Gaussian: $a_t, q_t, \Delta_t$ for group $i$ (left) and $j$ (right)

(b) German: $a_t, q_t, \Delta_t$ for group $i$ (left) and $j$ (right)

Figure 25: Error bar version of Fig. 7

(a) Uniform data when $r = 0$ and $T$ is large: Group $i$ (left), Group $j$ (right)

(b) Gaussian data when $r = 0$ and $T$ is large: Group $i$ (left), Group $j$ (right)

(c) German Credit data when $r = 0$ and $T$ is large: Group $i$ (left), Group $j$ (right)

(d) Dynamics when all data is human-annotated (i.e, $r = 1$) on synthetic datasets.

Figure 26: Error bar version of Fig. 13

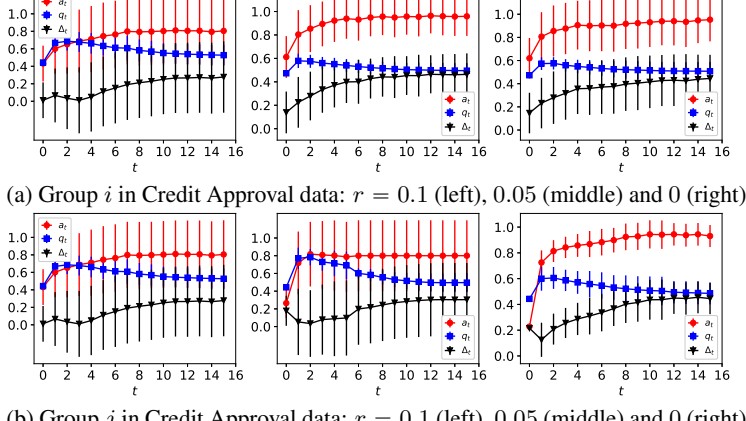

(a) Group $i$ in Credit Approval data: $r = 0.1$ (left), 0.05 (middle) and 0 (right)

(b) Group $j$ in Credit Approval data: $r = 0.1$ (left), 0.05 (middle) and 0 (right)

Figure 27: Error bar version of Fig. 11

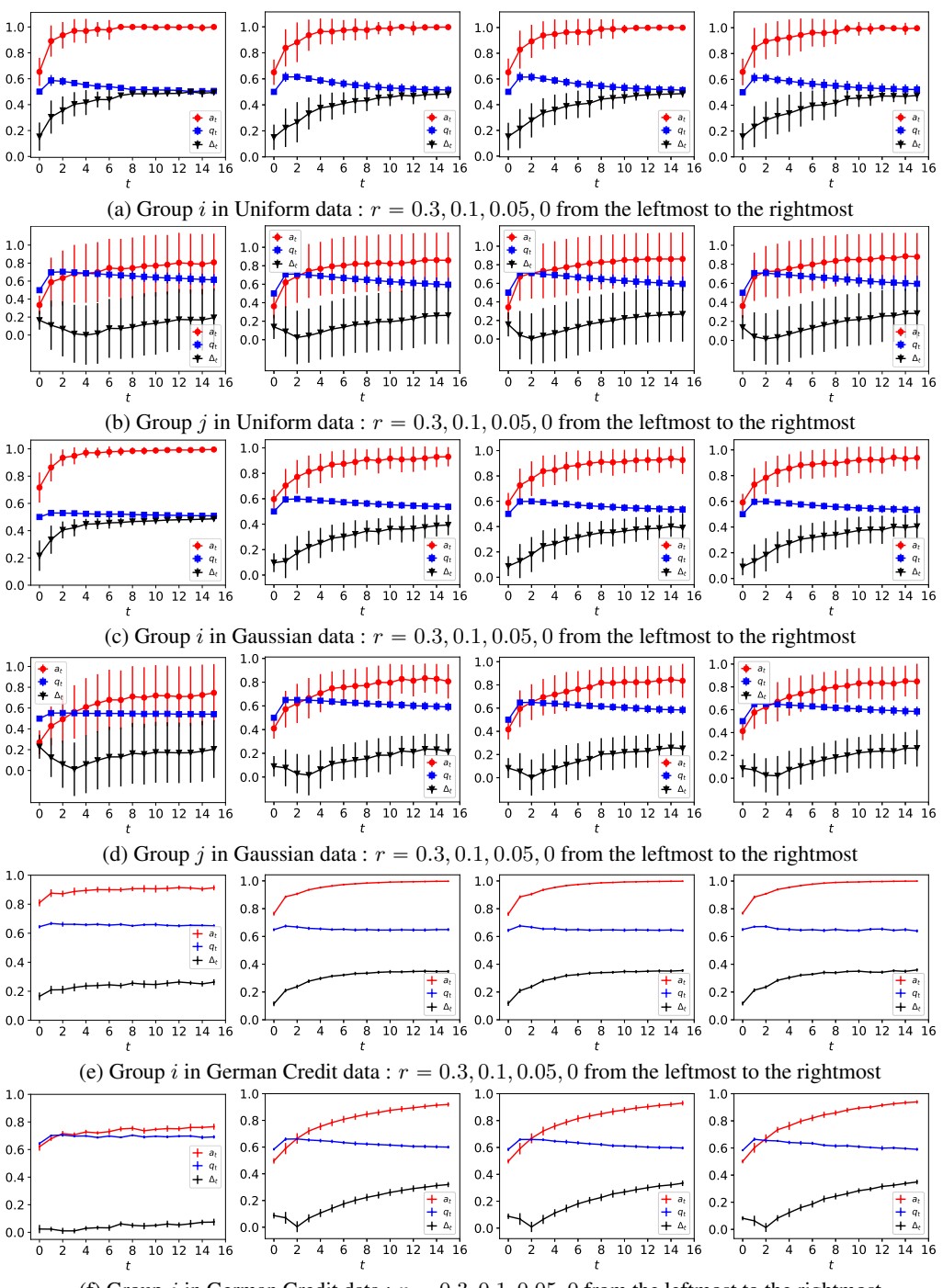

(a) Group $i$ in Uniform data : $r = 0.3, 0.1, 0.05, 0$ from the leftmost to the rightmost

(b) Group $j$ in Uniform data : $r = 0.3, 0.1, 0.05, 0$ from the leftmost to the rightmost

(c) Group $i$ in Gaussian data : $r = 0.3, 0.1, 0.05, 0$ from the leftmost to the rightmost

(d) Group $j$ in Gaussian data : $r = 0.3, 0.1, 0.05, 0$ from the leftmost to the rightmost

(e) Group $i$ in German Credit data : $r = 0.3, 0.1, 0.05, 0$ from the leftmost to the rightmost

(f) Group $j$ in German Credit data : $r = 0.3, 0.1, 0.05, 0$ from the leftmost to the rightmost

Figure 28: Error bar version of Fig. 15.

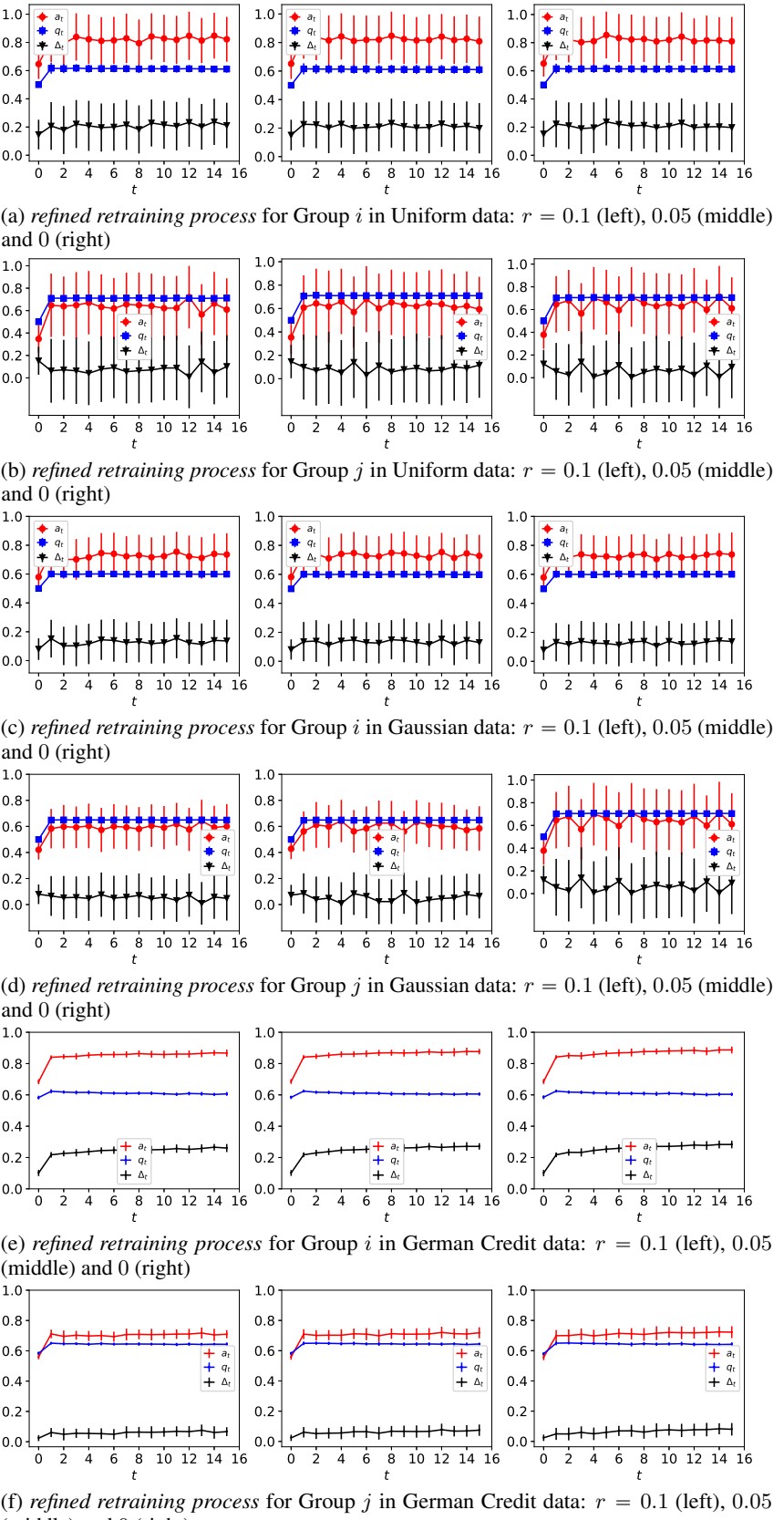

(a) *refined retraining process* for Group $i$ in Uniform data: $r = 0.1$ (left), 0.05 (middle) and 0 (right)

(b) *refined retraining process* for Group $j$ in Uniform data: $r = 0.1$ (left), 0.05 (middle) and 0 (right)

(c) *refined retraining process* for Group $i$ in Gaussian data: $r = 0.1$ (left), 0.05 (middle) and 0 (right)

(d) *refined retraining process* for Group $j$ in Gaussian data: $r = 0.1$ (left), 0.05 (middle) and 0 (right)

(e) *refined retraining process* for Group $i$ in German Credit data: $r = 0.1$ (left), 0.05 (middle) and 0 (right)

(f) *refined retraining process* for Group $j$ in German Credit data: $r = 0.1$ (left), 0.05 (middle) and 0 (right).

Figure 29: Error bar version of Fig. 16

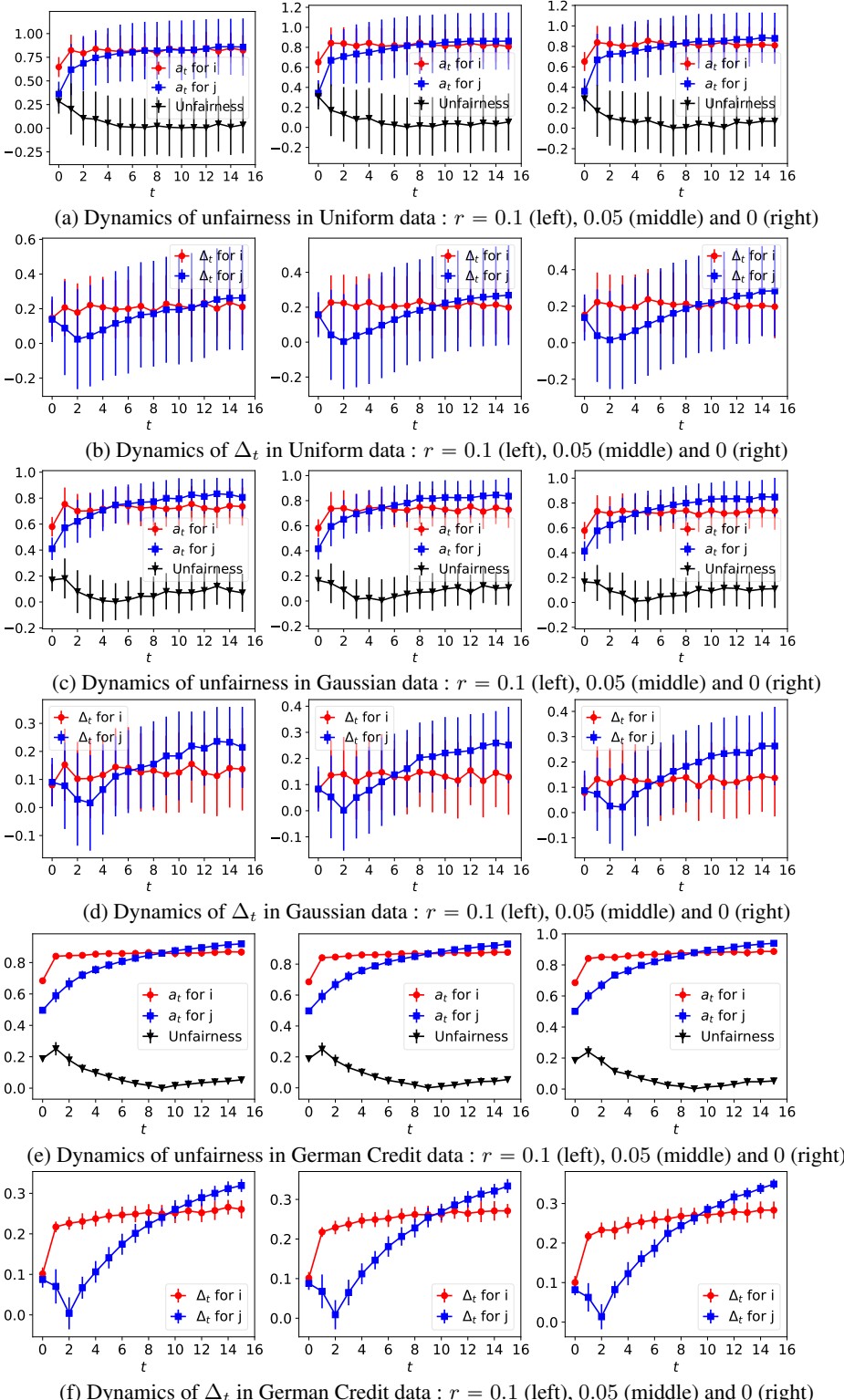

(a) Dynamics of unfairness in Uniform data : $r = 0.1$ (left), $0.05$ (middle) and $0$ (right)

(b) Dynamics of $\Delta_t$ in Uniform data : $r = 0.1$ (left), $0.05$ (middle) and $0$ (right)

(c) Dynamics of unfairness in Gaussian data : $r = 0.1$ (left), $0.05$ (middle) and $0$ (right)

(d) Dynamics of $\Delta_t$ in Gaussian data : $r = 0.1$ (left), $0.05$ (middle) and $0$ (right)

(e) Dynamics of unfairness in German Credit data : $r = 0.1$ (left), $0.05$ (middle) and $0$ (right)

(f) Dynamics of $\Delta_t$ in German Credit data : $r = 0.1$ (left), $0.05$ (middle) and $0$ (right)

Figure 30: Error bar version of Fig. 17

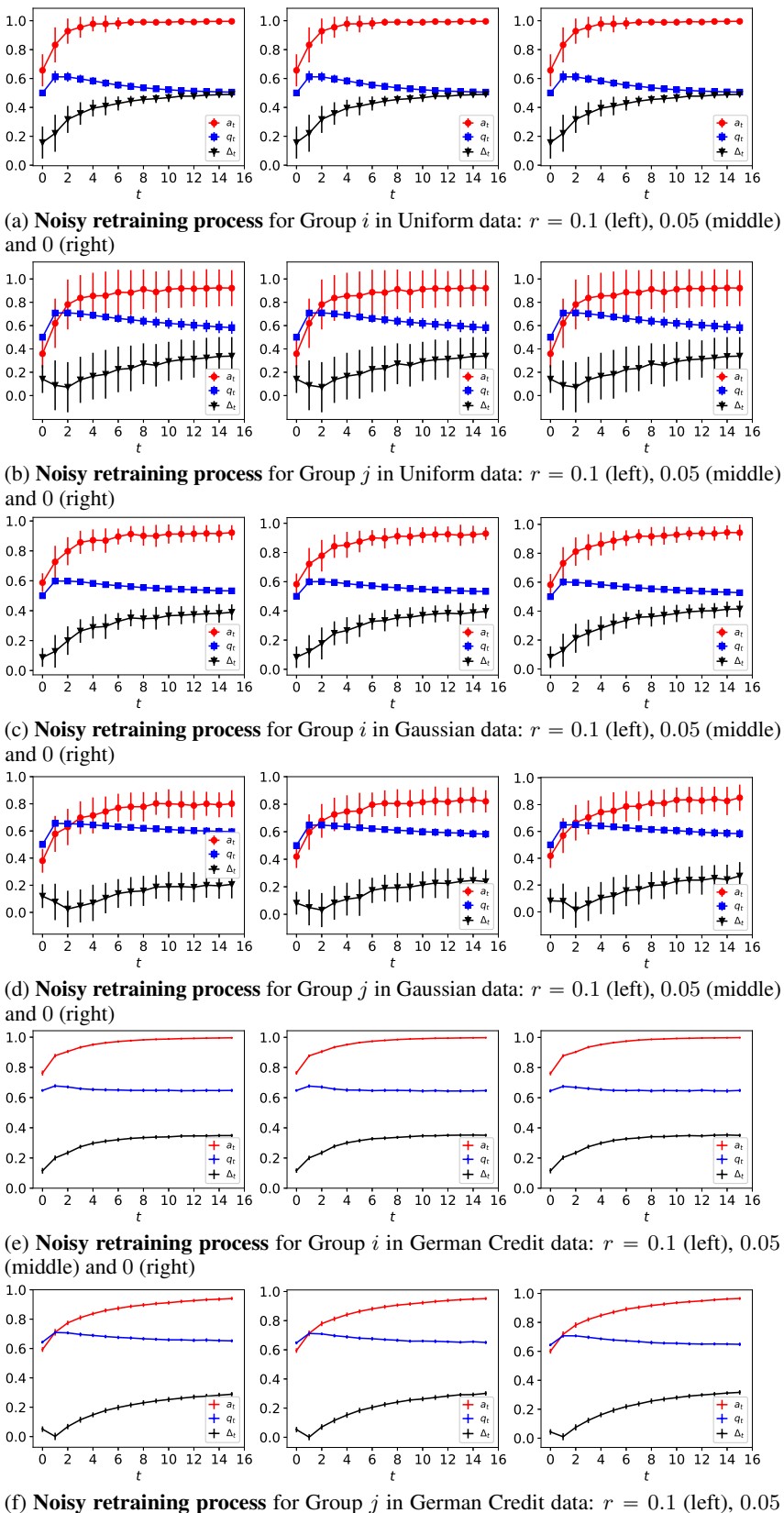

(a) **Noisy retraining process** for Group $i$ in Uniform data: $r = 0.1$ (left), 0.05 (middle) and 0 (right)

(b) **Noisy retraining process** for Group $j$ in Uniform data: $r = 0.1$ (left), 0.05 (middle) and 0 (right)

(c) **Noisy retraining process** for Group $i$ in Gaussian data: $r = 0.1$ (left), 0.05 (middle) and 0 (right)

(d) **Noisy retraining process** for Group $j$ in Gaussian data: $r = 0.1$ (left), 0.05 (middle) and 0 (right)

(e) **Noisy retraining process** for Group $i$ in German Credit data: $r = 0.1$ (left), 0.05 (middle) and 0 (right)

(f) **Noisy retraining process** for Group $j$ in German Credit data: $r = 0.1$ (left), 0.05 (middle) and 0 (right)

Figure 31: Error bar version of Fig. 18

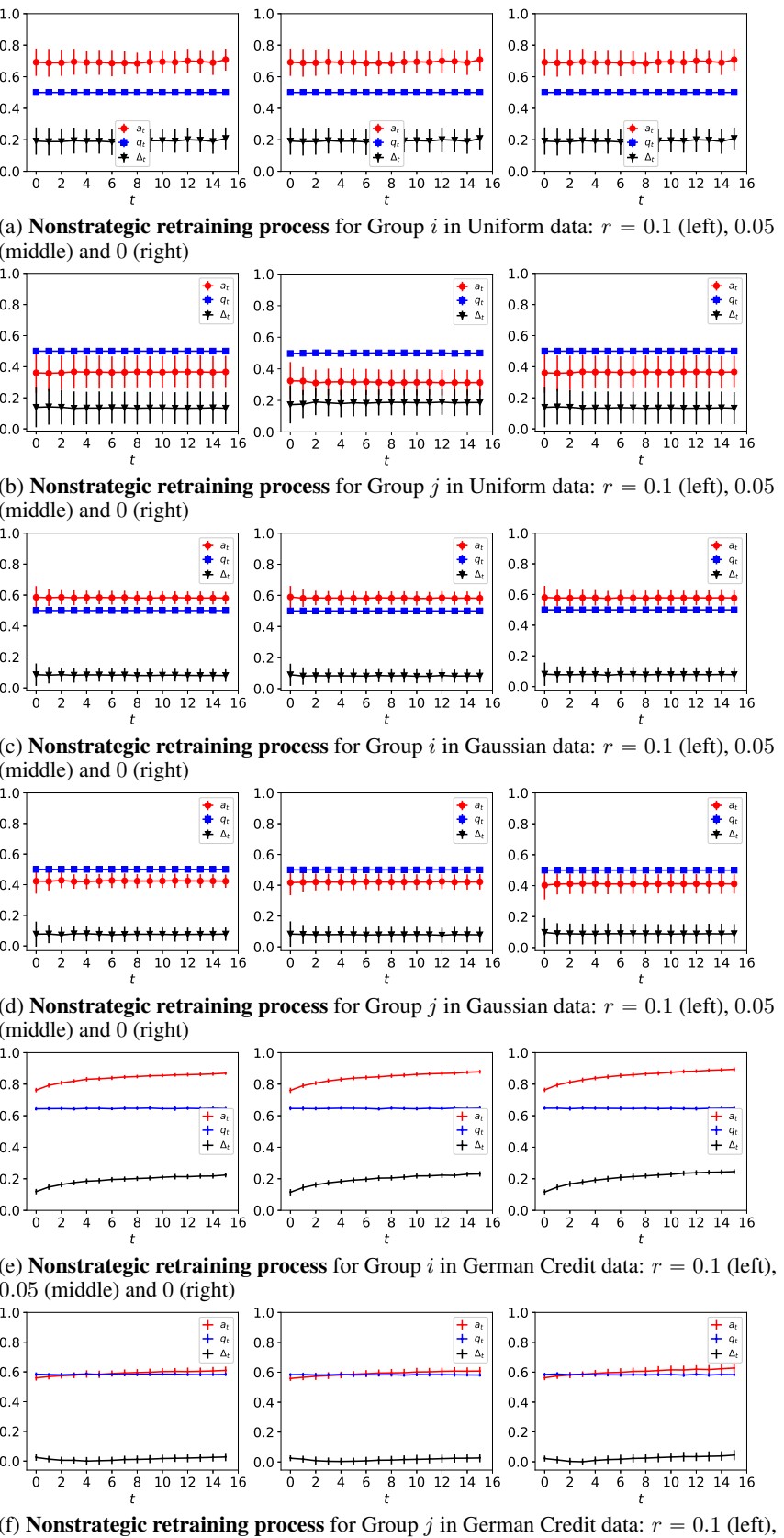

(a) **Nonstrategic retraining process** for Group $i$ in Uniform data: $r = 0.1$ (left), 0.05 (middle) and 0 (right)

(b) **Nonstrategic retraining process** for Group $j$ in Uniform data: $r = 0.1$ (left), 0.05 (middle) and 0 (right)

(c) **Nonstrategic retraining process** for Group $i$ in Gaussian data: $r = 0.1$ (left), 0.05 (middle) and 0 (right)

(d) **Nonstrategic retraining process** for Group $j$ in Gaussian data: $r = 0.1$ (left), 0.05 (middle) and 0 (right)

(e) **Nonstrategic retraining process** for Group $i$ in German Credit data: $r = 0.1$ (left), 0.05 (middle) and 0 (right)

(f) **Nonstrategic retraining process** for Group $j$ in German Credit data: $r = 0.1$ (left), 0.05 (middle) and 0 (right)

Figure 32: Error bar version of Fig. 19

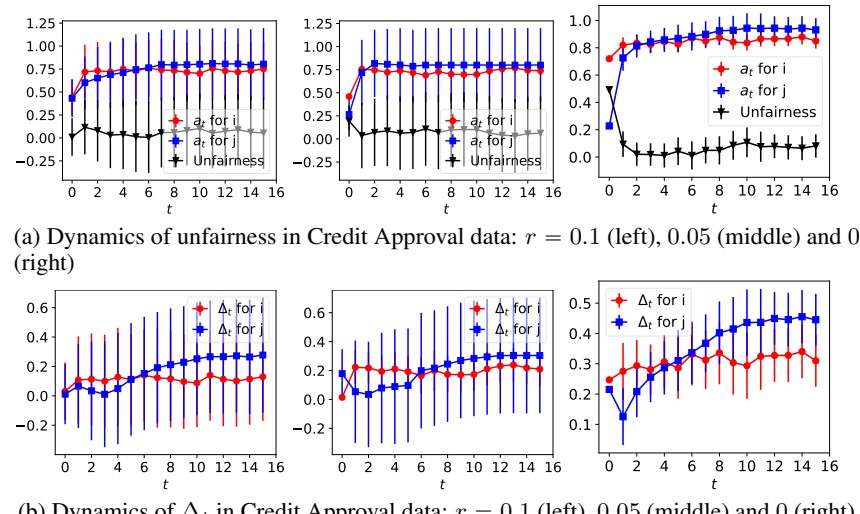

(a) Dynamics of unfairness in Credit Approval data: $r = 0.1$ (left), 0.05 (middle) and 0 (right)

(b) Dynamics of $\Delta_t$ in Credit Approval data: $r = 0.1$ (left), 0.05 (middle) and 0 (right)

Figure 33: Error bar version of Fig. 12

