# OpenReview forum: "Long-Term Impacts of Model Retraining with Strategic Feedback"
_ICLR.cc/2024/Conference — Submitted to ICLR 2024_

### Official Review · Reviewer_4hwy · 2023-10-31

**Soundness:** 3 good
**Presentation:** 2 fair
**Contribution:** 2 fair
**Rating:** 5
**Confidence:** 3

**Summary:**

This paper investigates acceptance rate, qualification rate, and classifier bias and unfairness when strategic agents interact with a machine learning system that retrains itself over time with both human-annotated and model-annotated samples. Due to the strategic interactions, it shows that the acceptance rate increases over time and gives conditions under which it can asymptotically reach 1. As expected, the qualification rate decreases over time. The disparity between these two metrics represents the classifier bias. The evolution of classifier bias is examined under different settings involving or not involving systematic bias. The results show that unless systematic bias is negative, classifier bias monotonically increases. To stabilize the dynamics of the variables above, the paper tweaks the generation of model-annotated samples by using a probabilistic sampler. It also proposes an early stopping mechanism to promote fairness when there exist two demographic groups with different sensitive attributes: one having a positive systematic bias and another having a negative systematic bias.

**Strengths:**

This paper studies an important problem faced in real-world ML deployments where agents who interact with the system are strategic and where both human and model-annotated samples are utilized. It provides an in-depth analysis of the acceptance rate, qualification rate, and classifier bias in such systems. The conclusions are intuitive but not surprising. The paper also touches upon the issue of fairness in the presence of disadvantaged groups and proposes a simple mechanism to balance it.

Experimental evaluation is extensive. Theoretical insights under simplified assumptions continue to hold under realistic settings in which some of the assumptions are violated.

**Weaknesses:**

To the best of my knowledge, it is the first time that strategic agents, repeated retraining, and a combination of human-annotated samples and model-annotated samples are investigated together in a study on the long-term impacts of model retraining. Most of the paper is concerned with building a theory that explains how biases grow over time and how unfairness can manifest itself. However, the most important question is how to prevent these biases and unfairness. The paper partly answers these questions by proposing a refined training process and an early stopping mechanism. They seem to be a small step towards the solution of the problem. Overall, this paper has a comprehensive formulation of the problem but an incremental solution.

While the paper investigates long-term interactions, it makes the simplifying assumption that the joint distribution $P_{XY}$ is fixed over time. This is clearly not the case, especially when who stays in the system or who comes into the system is determined by the previous interaction of the ML system with different demographic groups. User retention and the effect of retraining on the representation disparity seem to be neglected in the current work.

**Questions:**

- How Assumption 3.1 is justified? How does having a fixed hypothesis class allow us to ignore algorithmic bias? It clearly varies with $D^t$. Is it assumed that it does not vary much with $D^t$? Or is it assumed that the algorithmic bias is very small compared to other biases for all possible $D^t$ values? If so, can you justify this, perhaps by going over an example scenario and by providing experimental evidence?

- Theorem 3.5 does not shed light on the lower limit of the qualification rate. Can it get worse than the qualification rate under $P_{XY}$?

- If the decision-maker knows that there are long-term impacts, then why should it use a one-shot trained classifier like logistic regression? Isn’t it better to incorporate the knowledge of the long-term impacts of strategic agents and classifier-annotated samples within the loss function to be optimized?

- About implementation of the early stopping mechanism. Does the decision maker know that it has systematic bias? How does it know the identities of the advantaged and disadvantaged groups? If these are not known, then how can this early stopping mechanism be implemented?

- The appendix includes results where all examples are human-annotated. Investigating the figure there, can’t we conclude that it is better to forget about model-annotated samples and train only with human-annotated samples (although they are much fewer in number)?

---

> ### Author Response · Authors · 2023-11-18
> **Response to Reviewer 4hwy**
>
> Thanks for your comments. Please see our responses point by point.
>
> > Weakness 1: The paper is incremental in proposing solutions to biases and unfairness it uncovers.
>
> Answer:  The key of the section on fairness is to reveal the long-term fairness dynamics, where we discover that even when the systematic bias of human annotation always exists, simply retraining both groups for several rounds will lead to a fair classifier without any other explicit corrections. In the updated draft, we first provide an analysis of the long-term fairness dynamics under the retraining process when the two groups have different initial qualification rates (as suggested by Reviewer QsNX) and then analyze the dynamics of $a_t, q_t, \Delta_t$ while applying fairness intervention at each round on both original/refined retaining process. All details are reflected in Sec. 4 and App. D.2 of the updated draft.
>
> > Weakness 2: Simplifying assumption that the joint distribution $P_{XY}$ is fixed over time, neglecting user retention and representation disparity effects.
>
> Answer: In this paper, we mainly focus on retraining with different data sources and strategic agents, which is a novel setting that was never considered by previous literature. Adding participation dynamics may indeed model reality better, but it will add more complexity to the model, which may be a future direction.
>
> > Question 1: Justification for Assumption 3.1 regarding algorithmic bias and its variation with $D_t$
>
> Answer: When we consider a specific class of models, the algorithmic bias may not be negligible in practice, but the focus of our paper lies in the effects brought by the retraining process. Instead, algorithmic bias has already been extensively studied in the statistical learning theory and it is inevitable to some degree. With *model-annotated* samples, the sample complexity becomes very large, which already makes algorithmic bias as small as possible (the remaining bias is caused by the distribution itself). Our empirical results contain algorithmic bias, and it still demonstrates the validity of the theoretical results.
>
> > Question 2: lower limit in Thm 3.5
>
> Answer: Due to the monotonic likelihood assumption, the best response will not make agents' qualifications lower than $q_0$, so the lower limit will be $q_0$.
>
> > Question 3: why does the decision-maker use one-shot classifier?
>
> Answer: There are frameworks considering using repeated risk minimization to update a single classifier in long term [Perdomo et al., 2020]. However, the training strategy needs the loss function to satisfy rigorous assumptions and is not likely to converge to an optimal point. Thus, as a more spontaneous solution, the decision-maker tends to gather data each time to update the classifier, which is the common practice nowadays. Our paper aims to investigate whether this practice will bring unexpected outcomes.
>
> > Question 4: Implementation details about the early stopping mechanism in the presence of unknown systematic bias and group identities.
>
> Answer:  In practice, although the decision-maker may not know the exact magnitude of systematic bias, but it usually has sufficient prior knowledge of the group identities and the direction of the systematic bias. For example, in a college admission scenario, the admission may already have good knowledge on which ethnic group (race) is historically underrepresented, thereby assuming a negative systematic bias exists.
>
> > Question 5: Conclusion that training solely with human-annotated samples could be more beneficial?
>
> Answer: In our current experiment, we assume the decision-maker can obtain 2000 *human-annotated* samples each round, and the underlying distribution $P_{XY}$ is relatively easy to learn. In practice, the availability of *human-annotated* samples may be more limited, making the algorithmic bias significantly larger due to the low sample complexity.

---

### Official Review · Reviewer_Hjrr · 2023-10-31

**Soundness:** 3 good
**Presentation:** 3 good
**Contribution:** 2 fair
**Rating:** 3
**Confidence:** 4

**Summary:**

This paper studies strategic classification in a sequential framework where an ML model is periodically re-trained by a decision maker with both human and model-annotated samples. The paper analyzes how marginal quantities, such as the proportion of positively-labeled agents and the probability of positive classification change over time.

**Strengths:**

1. I commend the authors for finding and defining an interesting and novel problem! Not much work has focused on the interaction between strategic behavior and different model re-training strategies. The problem formulation is promising and well-written, though I do have some questions / issues with some of the details.

2. This work studies the interaction between a machine learning model and strategic human agents in a dynamic setting. Most previous works in strategic classification focus on a one-time deployment. Dynamic settings are of interest because realistically the human population evolves and the model is repeatedly updated.

3. Although the authors focus on a much more complicated setting, the question of what happens when agents are strategic and the model is updated using model-annotated samples (where the covariate distribution of the samples comes from the strategic agents’ covariates) is of independent interest. There has been some work related to this idea in the literature on repeated risk minimization in the performative prediction / strategic classification literature already, but to the best of my understanding, this particular question is still not completely understood. The framework that the authors propose can be used to make progress on this related problem.

**Weaknesses:**

1. The main weakness of the theoretical results in this work is that the authors focus on changes in ``marginal” quantities, such as the acceptance rate $E_{S_{t}}[A(f_{t}, P^{t})]$ instead of changes in conditional quantities, such as the acceptance rate conditional on a particular choice of covariates $x$, which is given by $E_{S^{t}}[f_{t}(X) \mid X=x].$ As a result, it is difficult to understand to what extent their results derive from the data provided from the strategic agents (covariate shift), the systemic bias in the human-annotated samples (one form of conditional shift), or the fact that the conditional distribution of the classifier $f_{t}(X) \mid X=x$ does not (second form of conditional shift). To what extent do the results derive from changes in the covariate distribution and changes in the classifier? In the absence of systemic bias, Theorem 3.3 seems to derive from the fact that after agents’ respond strategically to the model, they are more qualified to get a positive label (agents sampled from $P_{X}^{t}$ are more likely–based on P_{Y|X}-- to get a positive label than agents sampled from $P_{X}$). Thus the acceptance rate of agents would increase over time, simply because there are now more qualified agents. It would be helpful if the authors could clarify their presentation and discuss how much these different forces yield changes in the distribution of $f_{t}(X) \mid X=x$ the conditional distribution of the classifier.

2. This paper focuses on (1) human agents are reacting strategically to an ML model (2) retraining with human-annotated samples (3) the model is being updated with model-annotated samples. Studying these simultaneously, without discussion of what happens in the system when only one or two of these occur, makes it difficult to understand how different components of the system drive the main results. It might improve clarity for the authors to consider one of these at a time and describe what we expect to happen– for example, what would happen when human agents react strategically to a model over time? What would happen when human agents are strategic and the model is updated with model-annotated samples? This way the authors can build up to the entire complex system.


3. The assumption that agents’ covariate modification translates to genuine changes in the label is a somewhat strong assumption. This means that the classic “gaming” type behavior (where agents may manipulate their covariates while their labels remain fixed) is not permitted. This assumption greatly reduces the complexity of the problem because when “gaming” is not permitted, the strategic behavior problem becomes equivalent to covariate shift (no gaming essentially implies that $P_{Y|X}$ is fixed over time). Since this assumption is a departure from previous works in strategic classification, where previous works permit an agent to “game” and take meaningful actions that change their label [Kleinberg and Raghavan, 2020], the authors should flag it and make the contrast more clear in their paper. I noticed that the authors have a related work section in their appendix, but I would urge them to include some of these references in the main text of the paper.

4. It’s somewhat confusing that the covariate distributions of the model-annotated samples and the human-annotated samples are different. What’s the motivation for permitting the covariates of human-annotated samples to be sampled from the prior-best-response distribution? Given the dynamic nature of the setup, It seems more natural for both sets of samples to have the same covariate distribution (the post-best-respose covariate distribution).

Writing:
1. The paper could improve in notational clarity with regard to which quantities are finite/empirical and population level. For example the authors could name and express the distributions over the human-annotated samples, the model-annotated samples, and the previous training samples respectively. Then, the authors can define that the distribution over samples in the retraining dataset is a mixture distribution of the three components, where the weight on each component may differ depending on the proportion of each data type.

2. It might be helpful to use terms from the distribution shift literature to describe how different parts of the system affect the data distribution at time $t$. For example, in this work, the strategic behavior of the agents only results in covariate shift of the data distribution. The human-annotated samples represent a conditional shift of the original data distribution $P$. Meanwhile, the model-annotated samples represent a joint shift of the original data distribution, where the covariate shift can be attributed to the strategic behavior and the conditional shift arises from the distribution of $Y|X$ of the classifier being different from the $P$.

**Questions:**

1. It is not clear from the introduction how we should think about human-annotated samples. It would be helpful if the authors could clarify early on whether we view human-annotated samples to be reliable (the conditional distribution of the human-annotated samples is the same as $P_{Y|X}$ from the true distribution) or we believe the human-annotated samples to be inaccurate (the conditional distribution of the human-annotated samples does not have the same $P_{Y|X}$ as the true distribution). Later in the paper, it becomes clear that it is the latter. Furthermore, should we think of the human-annotated samples as being strategically supplied? Later in the paper, it seems like the human-annotated samples are not strategically supplied, because the covariates are drawn from the prior-best-response distribution and the labels are given by a systematically biased (but not necessarily strategic) decision maker. The confusion arises because the authors write that one of their motivating questions is – “how is the agent population reshaped over time when the model is retrained with strategic feedback?”

2. I'm especially interested in the setting where agents are strategic and the model is updated with only model-annotated samples? Will a fixed point arise [Frankel et al, 2022]? To what extent is this setting related to the repeated risk minimization in performative prediction [Brown et al, 2022, Perdomo et al, 2020]?

3. It would be helpful to see a discussion of the different retraining strategies, e.g. (1) using the strategic agents’ post-best-response covariate distribution labeled with model to retrain, (2) using the strategic agents’ post-best-response covariate distribution with ground-truth (accurate) labels to retrain, (3) using the prior-best-response covariate distribution (4) using i.i.d. data with ground-truth (accurate) labels to retrain.

Frankel, Alex, and Navin Kartik. "Improving information from manipulable data." Journal of the European Economic Association 20.1 (2022): 79-115.

Gavin Brown, Shlomi Hod, and Iden Kalemaj. Performative prediction in a stateful world. In Proceedings of the 25th International Conference on Artificial Intelligence and Statistics, pages 6045–6061, 2022.

Juan Perdomo, Tijana Zrnic, Celestine Mendler-Dünner, and Moritz Hardt. Performative prediction. In Proceedings of the 37th International Conference on Machine Learning, pages 7599–7609, 2020.

---

> ### Author Response · Authors · 2023-11-18
> **Response to Reviewer Hjrr**
>
> Thanks for the comments. Please see our responses point by point as follows.
>
> > Weakness 1: Focus on marginal quantities rather than conditional quantities makes results derivation unclear. It might improve clarity for the authors to consider one of these at a time and describe what we expect to happen.
>
> Answer: Firstly, our model mainly consists of 3 parts as shown in Eqn.(3) of the main paper: *human-annotated* samples, *model-annotated* samples, and agents' strategic best responses. We study them together since each of them is likely to play a role under the strategic retraining setting. Moreover, our model admits different proportions of *human-annotated* samples and *model-annotated* samples, which means the results are suitable for a process updated with only *model-annotated* samples or *human-annotated* samples. In App. G.3 (Lemma G.2), the proof details reveal that both $\mathbb{E}[f_t(X) | X = x]$ and $\overline{q}_t$ are increasing. However, note that the increasing $a_t$ is not due to the increasing $q_t$. As Prop. 3.4 shows, $q_t$ can decrease over time. It is just because the model "misannotate" some agents to be qualified.
>
> > Weakness 2: Assumption of covariate modification translating to genuine label changes is strong. How about gaming?
>
> Answer: Firstly, our work takes interest in how the qualification rate changes as the retraining goes on, so we assume the covariate shift because feature shifts will not produce a qualification rate change. Although the earliest works on strategic classification consider only the feature shifts, it is quite common for recent works to consider covariate shifts ([Rosenfeld, et al., 2020; Raab \& Liu, 2021; Guldogan et al., 2022]. Moreover, the works in causal strategic learning ([Miller et al., 2020]) suggest that the decision-maker can only consider causal features to ensure improvement.
>
> > Weakness 3: The covariate distributions for model-annotated and human-annotated samples are different without clear motivation.
>
> Answer: We assume the *human-annotated* samples are drawn from *prior-best-response* distribution $P_{XY}$ because the human annotation process is independent of the decision-making process.  At each round $t$, each human agent is classified by the model $f_t$ and best responds to $f_{t-1}$, the decision-maker never confuses the agents by simultaneously using human experts to label agents. Instead, human experts never participate in the interaction and human annotation is another process for the decision-maker to obtain information about the whole population by acquiring data from public datasets or some third parties, and then label them to recover the population distribution. This distribution is free of best response because the samples are not subject to any decision. To better illustrate this idea, consider an example as follows. In a loan application scenario, the bank obtains personal finance data from credit agency and other demographic data from the government to produce *human-annotated* data, while they keep using an automated decision model to reject or approve loan application and update this model with the aid of *human-annotated* samples. We also agree with the interesting nature of considering the *human-annotation* for *post-best-response* distribution and our model can incorporate this setting. Details are discussed in the general response.
>
> > Question 1: Clarity on how we should view human-annotated samples in terms of reliability and strategic supply.
>
> Answer: The model can either let *human-annotated* samples be reliable or not, but it is likely that they are not accurate and have a systematic bias due to several reasons (App. B gives examples). However, all results in Sec. 3.1 - 3.2 hold no matter whether systematic bias exists. This is why we do not state it until Sec. 3.3. The systematic bias exists not because of strategic supply. It exists mainly because of the complexity of labeling a sample under human-related decision-making scenarios. Sec. 4 and App. B illustrates this in detail.
>
> > Question 2: the setting where agents are strategic with only model-annotated samples.
>
> Answer: Our model can demonstrate this setting because the ratio between *model-annotated* and *human-annotated* samples is not constant. Specifically, Thm. 3.3, Prop. 3.4 will always hold when there are only *model-annotated* samples. Other results also need no modification.
>
> > Question 3: It would be helpful to see a discussion of the different retraining strategies
>
> Answer: thanks for this helpful suggestion. Just as the answer of the previous question, the ratio between *model-annotated* and *human-annotated* samples should not be a problem. For the *prior-best-response* and *post-best-response* strategies, we discuss it in the general response.

---

> > ### Author Response · Authors · 2023-11-18
> > **Continued Response to Reviewer Hjrr**
> >
> > > Question 4: Discussions of the mentioned related work
> >
> > Firstly, the line of work represented by [Kleinburg \& Raghavan, 2020] mainly focuses on designing an incentivizing mechanism for agents to invest their budget to improve themselves, which is a completely different topic to ours. Secondly, performative prediction [Perdomo et al, 2020] is able to handle strategic classification problem, but it is an iterative training procedure without retraining. Although the repeated risk minimization procedure can converge to a stable point under several strong assumptions, it can be still far from the optimal point, and the stable point does not have any practical explanation. Thus, it is more natural and commonly the case for a decision-maker to still keep retraining the model.

---

### Official Review · Reviewer_UjM1 · 2023-11-01

**Soundness:** 3 good
**Presentation:** 3 good
**Contribution:** 2 fair
**Rating:** 5
**Confidence:** 4

**Summary:**

The paper studies the dynamics between retrained machine learning models and strategic human agents who adapt their behavior over time. It finds that retraining makes models more likely to give positive decisions, but may reduce the proportion of agents with positive labels. To stabilize this, the paper proposes a method to improve algorithmic fairness. Experimental data and some theorems supports these findings.

**Strengths:**

The overall setting is interesting.

Theorem 3.3, 3.5, 3.6 seem correct and interesting. I spent about 1 hour per theorem. It's possible that mistakes exist but results seem plausible at a glance.

I think most of the value is in the theoretical results. The simulations are interesting, but are mostly window dressing for the mathematical formulation and ensuing theorems. I think it majority of the value in this paper is intellectual. With that being said, I am actually very fond of the formulation and I feel that the analysis of the emergent dynamics is extremely interesting.

**Weaknesses:**

My main issue is with Section 4.

The primary result of section 4, Theorem 4.1, seems like the authors are grasping at straws in order to have a Fairness section, rather than actually finding the most interesting emergent results in their model.

While social aspects of computing and ML are extremely timely and important issues, I fail to see the value of 4.1, given that the proposed "early stopping" is not particularly well-connected to a tangible use-case or real-world setting. The mathematical depth of 4.1 seems lackluster compared to Section 3.

The authors should consider removing Section 4. Instead, they could focus on removing convexity constraints in Thm 3.5. Convexity seems like a strong assumption -- it would be good to weaken it or further describe results in more general settings. I think that would strengthen this work more than a bolted-on fairness theorem. As can be seen in the supplement, there is seemingly little substance to theorem 4.1.

**Questions:**

Why is 4.1 interesting? Am I missing a practical application of this work? Or perhaps I am missing something interesting about the 4.1 from a theoretical perspective in terms of the depth of the result?

In rebuttal, I am looking for strong response. I think this paper could be as high as a 7 depending on the author response.

In fact, by simply REMOVING section 4, this paper is probably a 6. By replacing 4.1 with a substantive result, I think this could even be a 7.

I realize this is a bit of a reviewer bias in terms of fixating on one problematic theorem or claim in an otherwise sound paper, but in this case, an entire section of the paper is dedicated to what is essentially a meaningless result. I hope the authors can either explain what I am missing about section 4 or offer a replacement theorem in rebuttal.

---

> ### Author Response · Authors · 2023-11-18
> **Response to Reviewer UjM1**
>
> Thanks for your comments. Please see our responses point by point as follows.
>
> > Issues in Section 4
>
> Answer: The objective of section 4 is to study the effects on fairness solely caused by the retraining process with strategic agents, so we consider the setting where the two groups have no "innate" qualification rate difference (equal base rate). Thm 4.1 reveals that even when the systematic bias of human annotation always exists, simply retraining both groups for several rounds will lead to a fair classifier without any other explicit corrections. Regarding the questions on more substantive results on fairness, we can first provide an analysis of the long-term fairness dynamics under the retraining process when the two groups have different initial qualification rates (as suggested by Reviewer QsNX) and then analyze the dynamics of $a_t, q_t, \Delta_t$ while applying fairness intervention at each round on both original/refined retaining process. **All details are reflected in Sec. 4 and App. D.2 of the updated draft.**
>
> > Convexity in Thm 3.5
>
> The convexity of $F_X$ and $P_{Y|X}(1|x)$ will ensure that no matter where $f_t$ is and the cost matrix $B$ is, $q_t$ will always be decreasing. If any of the convexity constraint is not satisfied, $q_t$ will not be guaranteed to decrease. Consider an example on 1-d feature space where $P_X = 2 - 2x$, $P_{Y|X}(1|x) = x$, $B = 0.1$, and all new samples are model-annotated. Ideally, $f_0 = f_1 = \mathbf{1}(x \ge 0.5)$. Then agents having $x \in [0.4, 0.5)$ will best respond. However, at $t = 2$, since all agents are labeled 1 when $x \in [0.4, 0.5)$, $f_2$ is likely to set a threshold smaller than $0.5$. Assume the threshold is $0.5 - \theta$, then agents having $x \in [0.4 - \theta, 0.5 - \theta)$ will best respond. Note that $P_X$ is decreasing, so the improvement of agents at $t = 2$ will be larger than the improvement of agents at $t = 1$, and $q_t$ is not decreasing.
>
> However, if the conditions in Prop. 3.4 of the main paper is satisfied, $a_t$ will approach 1, this means $q_t$ will still eventually approach to $q_0$, which is the qualification rate of the *prior-best-response* distribution.

---

### Official Review · Reviewer_QsNX · 2023-11-01

**Soundness:** 2 fair
**Presentation:** 3 good
**Contribution:** 1 poor
**Rating:** 3
**Confidence:** 4

**Summary:**

The paper models and analyzes the dynamics of model retraining under strategic feedback. Specifically, how label prevalence and predicted prevalence change over time when agents' can adapt their features based on the previous time-step's model.

**Strengths:**

- The topic is interesting and warrants further studies.
- The paper is well written.
- The authors consider not only acceptance and qualification rates, but also how
fairness w.r.t. sensitive attributes evolves over time under this strategic
setting.

**Weaknesses:**

- Section on fairness is somewhat underwhelming and under-explored.
  - Assuming groups have equal base rates is entirely unrealistic.
  - If the purpose is to study "long-term impacts of model retraining", the proposed solution for fairness can't be "stop retraining the model".
  - Plus, assuming the decision-maker has some acceptance bias, and then relying on the same decision-maker to self-identify it and correct it, is not ideal.
  - It would be extremely interesting to have explored how strategic feedback (either individually or in the form of collective action) could've helped with improving fairness (and ideally using different base rates as a source of unfairness, which is still simplistic but already a great deal more realistic than a bias in acceptance rates alone).
  - The reason why early stopping can improve DP is because the retraining process (without "refinement") simply monotonically increases acceptance rate, which just showcases the modelling flaws.

- It's unclear to which degree the conclusions hold on any real-world scenario.
  - Of the two non-synthetic datasets ("German" and "credit approval"): (1) the
  experiments for "German" seem to have been ran on a synthetic version of the
  dataset as a result of fitting a KDE on the original data, and (2) the
  experiments on "credit approval" use only a small subset of features (two
  continuous features).
  - As there are a series of simplifying assumptions that are unlikely to hold
  in real-world settings, it seems important to verify to which degree violating
  these assumptions changes the conclusions.

- Shouldn't qualification rate for human-annotated samples also change as a
results of the same strategic behavior? How would an agent know whether it was
assigned to a model or human annotator?

**Questions:**

- Why does the qualification rate only change for the model-annotated samples
under strategic feedback, and not for human-annotated samples?
  - It's assumed that the "algorithm bias" is negligible; so whichever strategic
  feedback moves agents' features $X$ towards a more positive prediction by the
  classifier $f_{t-1}$ will also increase their qualification and acceptance
  rates for human annotators.
  - In other words, as the marginal feature distribution $P_X$ changes under
  strategic feedback, it would be expected that the qualification rate $Q(\mathcal{S}_{o,t-1})$
  would also change in response, no? Or are the human-annotated samples at each
  time step not drawn from the same distribution? And, if so, why not, and how
  could the decision-maker distinguish in practice whether to use a model or a
  human annotator?
  - I see that it's assumed that human annotated samples are drawn from the
  prior-best-response distribution, but how would (1.) the decision-maker
  distinguish these distributions, and (2.) the agents' know whether they were
  assigned to a model-annotator in order to know whether to employ strategic
  behavior.

---

> ### Author Response · Authors · 2023-11-18
> **Response to Reviewer QsNX**
>
> Thanks for your comments. We provide our response point by point as follows.
>
> > Weakness 1: Assuming groups have equal base rates is entirely unrealistic. It would be interesting to have explored how strategic feedback could've helped with improving fairness using different base rates.
>
> Answer: The assumption that groups have equal base rates is only used in the discussion of section 4 on fairness. In particular, we are concerned about how retraining with systematic bias under the strategic setting will influence fairness given the two groups are equally qualified and should be treated equally, so we make the equal base rate assumption. Furthermore, Thm 4.1 holds when group $i$ has a base rate larger than or equal to the one of group $j$ (not necessarily equal). For other situations under unequal base rates, we discuss it in detail in the general response to reviewers.
>
> > Weakness 2: The proposed solution for fairness can't be "stop retraining the model". And relying on the same decision-maker to self-identify and correct acceptance bias is not ideal
>
> Answer: The key of the section on fairness is to reveal the long-term fairness dynamics, where we discover that even when the systematic bias of human annotation always exists, simply retraining both groups for several rounds will lead to a fair classifier without any other explicit corrections. Instead of implementing the early stopping mechanism, we can consider a more common setting where a "hard fairness intervention" is performed at each round to only let the decision-maker choose from fair models. This is also discussed in detail in the general response.
>
> > Weakness 3: Early stopping can improve DP because the retraining process increases acceptance rate, showcasing modelling flaws.
>
> Answer: What we hope to show in Sec. 4 is that although the acceptance rate increases as the retraining process goes on, it is possible to have positive impact on fairness if we apply the early stopping mechanism.
>
> > Weakness 4: Dataset and real-world applications
>
> Answer: The theoretical model needs us to know $P_{XY}, P_X$. This is why we need to estimate the distribution through the KDE estimator. The common datasets for strategic learning settings are limited in samples, so many previous works have a protocol to generate samples. Moreover, except for the uniform synthetic dataset, all other datasets violate some of the assumptions in the main paper, and the results still verify the validity of our theoretical results.
>
> > Question 1: Why does the qualification rate only change for model-annotated samples under strategic feedback and not for human-annotated samples?
>
> Answer: We assume the *human-annotated* samples are drawn from *prior-best-response* distribution $P_{XY}$ because the human annotation process is independent of the decision-making process.  At each round $t$, each human agent is classified by the model $f_t$ and best respond to $f_{t-1}$, the *decision-maker* never confuses the agents by simultaneously using human experts to label agents. Instead, human experts never participate in the interaction and human annotation is another process for the decision-maker to obtain information about the whole population by acquiring data from public dataset or some third-parties, and then label them to recover the population distribution. This distribution is free of best response because the samples are not subject to any decision. To better illustrate this idea, consider an example as follows. In a loan application scenario, the bank obtains personal finance data from credit agency and other demographic data from government to produce *human-annotated* data, while they keep using an automated decision model to reject or approve loan application and update this model with the aid of *human-annotated* samples. We also agree with the interesting nature of considering the *human-annotation* for *post-best-response* distribution and our model can incorporate this setting. Details are discussed in the general response.

---

### Author Response · Authors · 2023-11-18
**General Response to Reviewers**

We thank the reviewers for their constructive comments. Regarding the questions in common, we updated our draft (both the main paper and appendix). We also provide a general response as follows.

## Discussions on Fairness (Section 4)

The objective of section 4 is to study the effects on fairness solely caused by the retraining process with strategic agents, so we consider the setting where the two groups have no "innate" qualification rate difference (equal base rate). Thm 4.1 reveals that even when the systematic bias of human annotation always exists, simply retraining both groups for several rounds will lead to a fair classifier without any other explicit corrections. Regarding the questions on more substantive results on fairness, we can first provide an analysis of the long-term fairness dynamics under the retraining process when the two groups have different initial qualification rates (as suggested by Reviewer QsNX) and then analyze the dynamics of $a_t, q_t, \Delta_t$ while applying fairness intervention at each round on both original/refined retaining process. **All details are reflected in Sec. 4 and App. D.2 of the updated draft.**

## Human-annotated Samples

We assume the *human-annotated* samples are drawn from *prior-best-response* distribution $P_{XY}$ because the human annotation process is independent of the decision-making process.  At each round $t$, each human agent is classified by the model $f_t$ and best respond to $f_{t-1}$, the *decision-maker* never confuses the agents by simultaneously using human experts to label agents. Instead, human experts never participate in the interaction and human annotation is another process for the decision-maker to obtain information about the whole population by acquiring data from public dataset or some third-parties, and then label them to recover the population distribution. This distribution is free of best response because the samples are not subject to any decision. To better illustrate this idea, consider an example as follows. In a loan application scenario, the bank obtains personal finance data from credit agency and other demographic data from government to produce *human-annotated* data, while they keep using an automated decision model to reject or approve loan application and update this model with the aid of *human-annotated* samples.

Regarding the reviewers' questions and suggestions, We also agree with the interesting nature of considering the *human-annotation* for *post-best-reponse* distribution and our model can incorporate this setting. Eq.(3) in Sec.3. **The details are in Sec.2 and App. D.1 of the updated draft**.

---

### Author Response · Authors · 2023-11-21

Thanks for the insightful comments. We hope our responses and updated manuscript address your concerns. As we approach the conclusion of the discussion period, your further review and feedback on our response would be highly appreciated.

---

### Meta-Review · Area_Chair_XZqj · 2023-12-06

**Metareview:**

The reviewers agree that paper has an interesting, novel and promising starting point with the idea of integrating within the same framework data generated by human annotators, data generated by AI models, and the long-term effects of retraining in the presence of strategic agents.

However, the reviewers raised concerns that couldn't be resolved regarding the presentation of the paper, mainly:
- it is not clear what role each component plays. While putting all these components together is certainly an interesting endeavour, it has been suggested that the authors start smaller and explain how each component changes the conclusions or social impacts.
- the fairness section has been identified as weaker than the rest of the paper and mostly useless. Maybe the authors could save the space for that section to expand on the main aspects of the framework.

During the discussion, we have explicitly raise the question of whether this paper in the current state was sufficiently novel to be accepted, understanding that authors cannot resolve all the details at the first submission of particularly new line of works. However, we came to the conclusion that the main components of the paper should be clarified and better motivated to meet ICLR's acceptance bar.

**Justification For Why Not Higher Score:**

We would have raised the score if there was more clarity on the interplay and role of each elements in the framework

**Justification For Why Not Lower Score:**

N/A

---

### Decision · Program_Chairs · 2024-01-16

Reject